

# 1 Impact of aerosols and clouds on decadal trends in all-sky
# 2 solar radiation over the Netherlands (1966 – 2015)

Reinout Boers[1], Theo Brandsma[1] and A. Pier Siebesma[1]
[1]KNMI, De Bilt, PO Box 201, Netherlands
*Correspondence to*: Reinout Boers (reinout.boers@knmi.nl)





**Abstract.** A 50-year hourly dataset of global shortwave radiation, cloudiness and visibility over the Netherlands
was used to quantify the contribution of aerosols and clouds to trends in all-sky radiation. The trend in all-sky
radiation was expressed as a linear combination of trends in fractional cloudiness, clear-sky radiation and cloud-
base radiation (radiation emanating from the bottom of clouds). All three trends were derived from the data
records. The results indicate that trends in all three components contribute significantly to the observed trend in
all-sky radiation. Trends (per decade) in fractional cloudiness, all-sky, clear-sky and cloud-base radiation were
respectively 0.0097±0.0062, 1.81±1.07 W m$^{-2}$, 2.78±0.50 W m$^{-2}$, and 3.43±1.17 W m$^{-2}$. Radiative transfer
calculations using the aerosol optical thickness derived from visibility observations indicate that Aerosol
Radiation Interaction (ARI) is a strong candidate to explain the upward trend in the clear-sky radiation. Aerosol
Cloud Interaction (ACI) may have some impact on cloud-base radiation, but it is suggested that decadal changes
in cloud thickness and synoptic scale changes in cloud amount also play an important role.
**1 Introduction**
Aerosols and clouds impact the solar radiation reaching the surface by radiative absorption and scattering.
Although there have been well-recorded trends in the all-sky radiation all over the globe it has been difficult to
precisely attribute such trends to trends in either aerosols or clouds. Wide-spread reductions in all-sky radiation
in the 1950 – 1970's ('dimming') have been followed by increases in later decades ('brightening'), especially in
Europe (Wild et al., 2005; Wild, 2009). Indeed, a thorough evaluation of all-sky radiation data over Europe
(Sanchez-Lorenzo et al., 2015) shows conclusively the distinct dip during the 1970's flanked on either side by
an earlier downward trend and a later upward trend. The later upward trends are thought to be the result of
regulatory restrictions on emissions of air pollution. Yet, modelling of this radiative effect (Allen et al., 2013) by
computing the impact of changing emissions of aerosols and aerosol precursors derived from CMIP5 have
shown that none of the 13 used models in that study can reproduce observational data.
One issue hampering the understanding of records of all-sky radiation is that the impacts of aerosols and clouds
need to be derived from a single record at observational sites where additional data for instance from clouds,
were often not present. This has led some investigators to group data into regions and rely either on cloud data
from stations in the immediate surroundings or from satellites (or both) to supplement their radiation records
(Norris and Wild, 2007). Even though good results on trends in clear-sky radiation can be obtained at sites
where direct and solar radiation are recorded at the same time such as Baseline Surface Radiation Network
stations (Long and Ackermann, 2000; Long et al., 2009; Wild et al., 2005; Gan et al., 2009), most often an
investigator will have to rely on single global radiation data records that are specific to the region of interest
(such as Manara et. al., 2016) or on data stored in the Global Energy Balance Archive (GEBA) archive. GEBA
data are of unmistakable quality but the peculiarities of the radiative signals typical to individual localities are
invariably lost in the abundance of data. It is therefore of great importance that regional studies are carried out
that record the changes in surface radiation in relation to atmospheric parameters that can influence such
changes.

In the context of Europe there have been a considerable number of regional studies that focus on trends in global
radiation and their attribution, such as in Germany (Liepert and Tegen, 2002; Liepert and Kukla, 1997; Liepert,



1997; Liepert, 2002)), in Germany and Switzerland combined (Ruckstuhl et al., 2008; Ruckstuhl and Norris,
2009; Ruckstuhl et al., 2010), in Estonia (Russak, 2009), in the general Baltic states (Ohvril et al., 2009), in
Spain ( Mateos et al., 2014), in Norway (Parding et al., 2014), northern Europe in general (Stjern et al, 2009)
and in Italy (Manara et al., 2015). Even though there are regional differences the summarized global or all-sky
radiation data from Europe combined (Sanchez-Lorenzo et al, 2015) displays a minimum in 1984 – 1985 at the
end of a 'dimming' period with a subsequent return to higher values. The consensus about the decadal trends in
global radiation hides a considerable discourse about the attribution of the radiation trends. Of the parameters of
interest when investigating the trends in all-sky radiation namely clear-sky radiation, cloudy-sky radiation and
fractional cloudiness, the first two have been difficult to isolate from data and were addressed in only a few
studies (Wild, 2010). Yet an increasing number of studies indicate that there are good reasons to believe that
Aerosol Radiation Interaction (ARI) is responsible for the rise in all-sky radiation after 1985 (f.e. Philipona et al,
2009; Manara et al, 2016; Ruckstuhl et al, 2008) although the timing of the minimum or intensity cannot be
simulated very well using current aerosol emission inventories (Ruckstuhl and Norris, 2009; Liepert and Tegen,
2002, Romanou et al, 2007; Turnstock et al, 2015). About the influence of clouds, the situation continues to be
elusive. While it is obvious that clouds are important, the difficulty here is that there are several factors that
control their impact. For example there are considerable regional differences in fractional cloudiness (Norris,
2005): fractional cloudiness is constant in Northern Europe (Parding et al, 2014), in Germany before 1997
(Liepert, 1997) well after the minimum in global radiation in 1984, and  is declining in the period after 1997 in
Switzerland and Germany, at least up to 2010 (Ruckstuhl et al, 2010).  Furthermore, cloud optical thickness
changes can be the result of changes in microphysics or cloud thickness and current observations are not able to
separate the two effects. Nevertheless, modelling and observation studies by Romanou et al (2007), Ruckstuhl
and Norris (2009), Chiacchio and Wild (2010), Liepert (1997) and Liepert and Kukla (1997) suggest a definite
but mixed  role for dynamical as well as microphysical influences impacting the trend in all-sky radiation.

Attribution studies using only surface-based observations must rely on supplemental data, namely those of
clouds (predominantly fractional cloudiness) and aerosols. Data on fractional cloudiness are mostly collected
simultaneously with radiation data. Up to the mid-1990 clouds were observed by human observers but since
then the role of the observers is taken over by  ceilometers.  Apart from occasional sun photometer records
(Ruckstuhl et al (2008) data on aerosol are often unavailable. However, recent studies by Wu et al. (2014) and
Boers et al. (2015)  have shown that it is possible to retrieve useful aerosol optical thickness data from surface
visibility records. The principal idea behind both studies is almost 50 years old (Eltermann, 1970; Kriebel, 1978;
Peterson and Fee, 1981; and revived by the work of Wang, 2009) and asserts that clear-sky optical thickness is
most often caused by aerosols residing in the planetary boundary layer which can be characterized by the optical
extinction at 550 nm. This parameter is by definition proportional to the inverse of atmospheric horizontal
visibility which in turn is a quantity abundantly observed over at least 50 years, often together with observations
of radiation.

Because of the importance attached to potential attribution of observed regional trends in all-sky radiation to
aerosols and / or clouds, we analyze hourly records of radiation, cloudiness and visibility data at five climate
stations in the Netherlands for the 50-year period 1966–2015. The two aims of this study are a) to quantify the





decomposition of the all-sky flux into its contributing components and compute the decadal trends in the
components, and b) to discern the relative importance of aerosols and clouds in shaping the observed trends.

The remainder of this paper is organized as follows: Section 2 presents the theory and analysis procedures to
obtain clear and cloudy-sky signals from the all-sky data. The procedures combine radiation and cloud coverage
data. Equations are derived describing the manner in which the all-sky radiation is explicitly dependent upon
fractional cloudiness, clear-sky radiation and radiation emanating at cloud-base. The equations are based on
elementary principles but we believe that this is the first time that these dependencies are explicitly quantified,
although the work by Liepert (1997), Liepert (2002), Liepert and Kukla (2002), and Ruckstuhl et al. (2010)
contain elements similar to our work.

In section 3 the data analysis is discussed: all meta-data for all stations recorded between the late 1950's and
today were examined in order to better understand the impact of any changes in instruments and location and
calibrations on the data. Homogeneity tests were performed to discern any possible discontinuities in the data
and to understand whether all climate stations indeed belonged to the same climatological regime. Also attention
is given to a break in the cloud observations that occurred in 2002 with the transition from the human observer
to the ceilometer. Section 4 show the results. The relative influence of clear-sky radiation, cloudy-sky radiation
and fractional cloudiness on the all-sky radiation are shown. Also the relative merits of ARI and ACI in
influencing the all-sky radiation are discussed.

Section 5 concludes this paper with discussion and conclusions.
**2 Method**
**2.1 Decomposition of all-sky radiation into clear and cloudy sky components**
An important aspect of this paper is to quantify the various radiative contributions to the all-sky radiation. It is
shown in this subsection that there is an elegant way to do so while invoking a minimum set of assumptions.
The radiative contributions arise from skies under clear, partly cloudy or overcast sky conditions. The presence
of cloud cover which is recorded simultaneously with the radiation assures that it is possible to quantify these
different contributions. Cloud cover is normally recorded in oktas (0-8) so that nine different contributions to the
radiation can be identified, which together build up the all-sky radiation.

For each okta value it will be assumed that the observed radiation is a linear combination of clear-sky radiation
and radiation emanating from cloud-base, each with cloud fraction weight factors that correspond to the okta
value at hand. The result is an equation which casts the all-sky radiation as a function of only three components:
1) the clear-sky radiation, 2) the cloud-base radiation and 3) the fractional cloudiness. The process to calculate
the three components will be repeated for each year in the period 1966 – 2015, resulting in three time series.
The method thus assures that the relative importance of clear-sky radiation, cloud-base radiation and fractional
cloudiness to the trend in all-sky radiation can be quantified.





We analyze the trends of time series of global radiation $S(y_k)$ where $S$ is the yearly averaged global radiation, $y_k$
is a year in the period 1966 – 2015 and k is the index of the year. We write $S(y_k)$ as a function of two controlling
variables: fractional cloudiness ($c$) and cosine of solar zenith angle ($\mu_0 = cos(\theta_0)$). Each of these two
parameters varies between 0 and 1 (i.e. when the sun is below the horizon the variable $\mu_0$ is set to zero).

In the observations from meteorological stations the global radiation comes in discrete values, in our case as
hourly averages, 8760 or 8784 values in a year. Each of these hourly averages is thus assigned a specific value
of $\mu_0$. The index i is the bin index of counting over $\mu_0$. To build up the probability space for $\mu_0$ bins of $\mu_0$ can be
selected (for example with width 0.05).

Observations of cloudiness are usually assigned in oktas. Okta values (0 – 8) are associated with specific
margins of fractional cloud coverage (see table 1 of Boers et al, 2010). We will designate the fractional
cloudiness associated with each okta value as $c_j$ where j = 0 – 8. The bivariate distribution function can then be
constructed as
$$p(\mu_0 = \mu_{0ik}, c = c_{jk}) = \frac{N_{ijk}}{N_k}$$    (1a)
where $N_{ijk}$ is the number of observations in a single bin and
$$\sum_i \sum_j N_{ijk} = N_k \quad \text{and} \quad \sum_i \sum_j p(\mu_0 = \mu_{0ik}, c = c_{jk}) = 1$$    (1b,c)
Marginal distribution functions of Eq. (1) are
$$f_c(c_{jk}) = \sum_i p(\mu_0 = \mu_{0ik}, c = c_{jk}) = \frac{\sum_i N_{ijk}}{N_k} = \frac{N_{jk}}{N_k}$$    (2)
where $f_c(c_{jk})$ is the fractional occurrence of cloud cover within a specific okta value, and
$$f_{\mu_0}(\mu_{0ik}) = \sum_j p(\mu_0 = \mu_{0ik}, c = c_{jk}) = \frac{\sum_j N_{ijk}}{N_k} = \frac{N_{ik}}{N_k}$$    (3)
where $f_{\mu_0}(\mu_{0ik})$ is the distribution of cosines of solar zenith angle. While the distribution $f_{\mu_0}(\mu_{0ik})$ is
invariant with time as it is solely dependent on the latitude of the observations, $f_c(c_{jk})$ is varying with time due
to yearly and possible decadal trends. Yearly averaged fractional cloudiness $c(y_k)$ is found as the expected value
of $c$ of the marginal distribution $p_c$
$$c(y_k) = \sum_{j=1}^{8} c_{jk} f_c(c_{jk})$$    (4)
The yearly averages $S(y_k)$ can be computed as the expected value of $S$, namely the double summation over all
values of $c$ and $\mu_0$ that jointly occur in a single year
$$S(y_k) = \sum_i \sum_j S(\mu_0 = \mu_{0ik}, c = c_{jk}) p(\mu_0 = \mu_{0ik}, c = c_{jk})$$    (5)
Here $S(\mu_0 = \mu_{0ik}, c = c_{jk})$ is the average value of $S_k$ in the bin (i,j,k).





For each okta class we can derive the distribution of zenith angles as the conditionally sampled bivariate
distribution at the specific okta class $c_{jk}$:
$$f_{\mu_o}(\mu_0 = \mu_{0ik} \mid c = c_{jk}) = \frac{p(\mu_0 = \mu_{0ik}, c = c_{jk})}{f_c(c_{jk})}$$ (6)
We now obtain the yearly averaged global radiation in each okta class as the expected value of the hourly global
radiation data sampled conditionally with okta class:
$$S_{c_j}(y_k) = \sum_i S(\mu_0 = \mu_{0ik}, c = c_{jk}) f_{\mu_0}(\mu_0 = \mu_{0ik} \mid c = c_{jk})$$ (7)
Combining Eq. (5), (6) and (7) yields
$$S(y_k) = \sum_j f_c(c_{jk}) S_{c_j}(y_k)$$ (8)
Provided that there are adequate observations of cloudiness to select each observation of global radiation
according to the okta class in which it occurs, it is possible to calculate $S_{c_j}(y_k)$ directly from the observations.

The assumption we make at this point is that
$$S_{c_j}(y_k) = (1 - c_{jk}) S_{c_0}(y_k) + c_{jk} S_{cb,c_j}(y_k)$$ (9)
where $S_{cb}$ is the cloud-base radiation. Although Eq. (9) is a customary approximation, it is almost certainly
incomplete as it neglects possible contributions to the flux from three-dimensional photon scattering between
clouds, in particular when cloud cover is broken. However, to our knowledge no useful correction to Eq. (9) has
been published taking such scattering into account. Eq. (9) provides the means to estimate cloud-base radiation
as all other parameters are known. Inserting Eq. (9) into Eq. (8) with some manipulation and using the definition
of Eq. (4) yields the desired result:
$$S(y_k) = S_{c_0}(y_k)[1 - \sum_{j=1}^{8} f_c(c_{jk}) c_{jk}] + \sum_{j=1}^{8} f_c(c_{jk}) c_{jk} S_{cb,c_j}(y_k) =$$
$$= S_{c_0}(y_k)[1 - c(y_k)] + c(y_k) S_{cloud}(y_k)$$ (10)

where
$$S_{cloud}(y_k) = \frac{\sum_{j=1}^{8} f_c(c_{jk}) c_{jk} S_{cb,c_j}(y_k)}{\sum_{j=1}^{8} f_c(c_{jk}) c_{jk}}$$ (11)
The parameter $S_{cloud}(y_k)$ is thus the cloud fraction weighted cloud-base radiation. Eq. (10) quantifies the all-
sky radiation as a function of three variables: namely the clear sky radiation, the weighted cloud-base radiation
and the fractional cloudiness.





**2.3 Proxy radiation**
It has long been recognized that $S_{c_j}(y_k)$ has large year-to-year fluctuations because $p(\mu_0 = \mu_{0ik}, c = c_{jk})$
varies from year-to-year. Extended periods of cloudiness of certain types that influence $p(\mu_0 = \mu_{0ik}, c = c_{jk})$
are associated with synoptic systems that may occur randomly during the year. This means that trend analysis
based on Eq. (7) is subject to large uncertainties that can only be alleviated by collecting data over large areas so
that different synoptic systems are sampled at the same time (Liepert, 2002), or by averaging $S_{c_j}(y_k)$ over
several years and then performing trend analysis on the reduced and averaged data set (Liepert and Tegen,
2002)). Over a relatively small region as the Netherlands Eq. (7) is unsuitable to use. In fact Ruckstuhl et al
(2010) demonstrated that the use of the radiation data in its pure form would lead to wrong interpretations of
trends. To reduce the uncertainty in estimates of $S_{c_j}(y_k)$, in particular when estimating the global radiation
under cloudless skies $S_{c_0}(y_k)$ some investigators have resorted to fitting an 'umbrella' function of clear-sky
radiation over all observations within one year (Long et al, 2009; Ruckstuhl et al, 2010) based for example on
discrimination of clear skies by analysis of direct and diffuse radiation. In our formulation the approach of
fitting an umbrella function is equivalent to a procedure whereby $S(\mu_0 = \mu_{0ik}|c = c_{0k})$ is fitted by a function
$G_{c_{0k}}(\mu_{0ik})$. When we proceed in this way, the parameter $S_{p,c_0}(y_k)$ which is a proxy for $S_{c_0}(y_k)$ is
calculated as
$$S_{p,c_0}(y_k) = \sum_i G_{c_{0k}}(\mu_{0ik}) f_{\mu_0}(\mu_{0ik}) \qquad (12)$$
This is based on strong theoretical arguments to suggesting that $G_{c_{0k}}(\mu_{0ik})$ is a monotonically increasing
function of $\mu_{0ik}$ given a specific value of $c_j$. The use of the marginal distribution $f_{\mu_0}(\mu_{0ik})$ in the summation
assures that the entire distribution of cosines of solar zenith angles representative for the location at hand is used
in the calculation rather than conditional distribution $f_{\mu_0}(\mu_0 = \mu_{0ik}|c = c_{0k})$ which is highly variable from
year-to-year and for which only a summation over a limited set of observations can be used.

In this paper the approach will be to generalize Eq. (12) to all nine okta values as
$$S_{p,c_j}(y_k) = \sum_i G_{c_jk}(\mu_{0ik}) f_{\mu_0}(\mu_{0ik}) \qquad (13).$$
In other words we will calculate functions of the type $G_{c_jk}(\mu_{0ik})$ for each okta value using the observations at
hand.

The notion that the functions $G_{c_jk}(\mu_{0ik})$ are monotonic increasing with $\mu_{0ik}$ comes from Beer's Law stating that
for a single wavelength only the optical thickness of the atmosphere and $\mu_{0ik}$ itself are parameters controlling the
change in downwelling radiation with $\mu_{0ik}$
$$S_s = \mu_0 S_e \exp(-\tau / \mu_0) \qquad (14)$$





Here $S_s$ is the downwelling radiation at the surface, $S_e$ is the extraterrestrial radiation, and $\tau$ is the optical
thickness of the atmosphere.

Even though the global radiation is a wavelength-integrated quantity, the scattering through the atmosphere
consisting of water droplets, ice crystals and aerosols at high relative humidity can in first order be assumed to
be conservative. Therefore, it is reasonable to assume that $G_{c_{jk}}(\mu_{0ik})$ has a functional form resembling Eq.
(14). When regressed through data taken over an entire year the fitted line has a parameter akin to the yearly
averaged optical thickness of the atmosphere as its sole controlling variable.

Consequently, we will adopt the function
$$G(\mu_0) = \mu_0 A \exp(-B/\mu_0) \qquad (15)$$
where B is a parameter depending on $\mu_0$ according to
$$B(\mu_0) = \alpha \mu_0^{\beta} \qquad (16)$$
as the diffuse radiation arriving at the surface is weakly dependent upon $\mu_0$.

The year-to-year determination of proxies in Eq. (13) is used in this paper as it will yield more stable results
than the determination of global radiation using the original Eq. (8). The approach will avoid all seasonal
elements and yearly variations that are inherent in the distribution $f_{\mu_0}(\mu_0 = \mu_{0ik}|c = c_{jk})$ due to the yearly
variable numbers of $\mu_{0ik}$ values necessary to compute the conditionally sampled data. Therefore, the computed
trends of proxies will reflect the yearly changing transmission through the atmosphere, which is the purpose of
this study.

Parallel to Eq. (10) we can write for the proxy global radiation
$$S_p(y_k) = S_{p,c_0}(y_k)[1 - c(y_k)] + c(y_k)S_{p,cloud}(y_k) \qquad (17)$$
where $S_{p,cloud}(y_k)$ is obtained from an equation identical to Eq. (11) with $S_{cb}(y_k)$ replaced by $S_{p,cb}(y_k)$.
In summary, the parameters $S_{p,c_0}(y_k), S_{p,cb}(y_k), S_{p,cloud}(y_k)$ are obtained from the proxy analysis in Eqs.
(12) – (16). However, note that $S(y_k) \neq S_p(y_k)$ as the proxy analysis is based on an evaluation of proxy
fluxes, not of the 'real' fluxes. In the analysis to be performed, however, differences between them turned out to
be less than 5%.

**2.4 Analysis of trend**
Once a time series of proxy radiation values is obtained it is possible to compute trends. As explained in the
previous section trends in the observed time series of clear-sky and cloudy sky radiation are not very useful due
to the year-to-year variability. However, trends in the proxy radiation time series do not suffer from such noise
and thus can yield meaningful results. A single equation will be derived for the trend in all-sky (proxy) radiation





from which is emerges that such trend is the result of three components: a) a trend in fractional cloudiness, b) a
trend in clear sky radiation and c) a trend in radiation at cloud-base.

To derive trends from the yearly averages (proxy) data we write:
$c(y_k) = \overline{c} + c'(y_k)$, $S_{p,c_0}(y_k) = \overline{S_{p,c_0}} + S'_{p,c_0}(y_k)$, $S_p(y_k) = \overline{S_p} + S'_p(y_k)$,
$S_{p,cloud}(y_k) = \overline{S_{p,cloud}} + S'_{p,cloud}(y_k)$                                                      (19)

Here the bar represents an average over 5 decades of the yearly averages, and the primed variables are the yearly
deviations from the decadal averages. Inserting into Eq. (17) yields

$S_p(y_k) = \overline{S_p} + S'_p(y_k) = (1 - \overline{c} - c'(y_k))(\overline{S_{p,c_0}} + S'_{p,c_0}(y_k)) + (\overline{c} + c'(y_k))(\overline{S_{p,cloud}} + S'_{p,cloud}(y_k))$

260                                                                                                                 (20)

Defining     $\overline{S_p} = (1 - \overline{c})\overline{S_{p,c_0}} + \overline{c}\,\overline{S_{p,cloud}}$     and collecting terms yields
$S'(y_k) = c'(y_k)(\overline{S_{p,cloud}} - \overline{S_{p,c_0}}) + (1 - \overline{c})S'_{p,c_0}(y_k) + \overline{c}S'_{p,cloud}(y_k) + c'(y_k)(S'_{cloud}(y_k) - S'_{p,c_0}(y_k))$

263                                                                                                                 (21)

Eq. (21) is the desired result. The first component on the right hand side represents perturbations / trend in
fractional cloudiness multiplied by the difference in cloud-base and clear-sky radiation, which is negative.
Therefore positive trends in fractional cloudiness will impact as a negative trend component in building up the
all-sky radiation. The second term represents the clear-sky perturbations / trend weighted by the average
occurrence of clear skies (in our case approximately 0.32). The third term represents the perturbations / trend in
cloud-base radiation weighted by the fractional cloud cover (in our case approximately 0.68). A fourth term not
shown here is a cross correlation term which in practice can be neglected.

Eq. (21) explains to a large extent the difficulties in attribution studies of the all-sky radiation. Not only the
trends in fractional cloudiness, clear-sky and cloud-base radiation are important, but also their relative weight as
determined by the mean fractional cloudiness and the difference between the mean clear-sky and cloud-base
radiation. In other words, there are a total of five different factors contributing to the trend in all-sky radiation.
For example, when the mean cloud fraction is large, as in northwestern Europe, the impact of the trend in clear-
sky radiation on the trend in all-sky radiation will be relatively modest in comparison to the impact of trend in
cloud-base radiation. The latter would be weighted by a factor 2 (0.32 versus 0.68) more than the trend in clear-
sky radiation.
**2.5 Retrieval of aerosol optical thickness**
Once the method to decompose the all-sky radiation into its clear-sky and cloudy-sky (proxy) components has
been applied and a trend analysis is performed, then it is our goal to seek an answer to the question which
processes might be responsible for their long-term change. Although possible long-term changes in the synoptic
conditions are a conceivable influence an obvious candidate for exploration of cause is the changing aerosol



content of the atmosphere. Aerosol content / concentration was not directly observed but visibility was recorded
throughout the period from which aerosol optical thickness was derived.

Aerosol optical thickness is the single most controlling factor in changing clear-sky radiation. A radiative
transfer model is used here to calculate the clear-sky radiation as a function of the changing optical thickness.
The output was compared to the observed clear-sky radiation. The process whereby aerosol can directly affect
clear-sky radiation is denoted as the aerosol direct effect or, using a term used in the IPCC (IPCC, 2013) report,
the Aerosol Radiation Interaction (ARI).

Aerosols can also affect the microphysical structure of clouds which in turn affects its radiative structure, a
process which is commonly denoted as the aerosol indirect effect, or Aerosol Cloud Interaction (ACI, as using
the terminology of IPCC, 2013).

Aerosol optical thickness ($\tau_a$) is a function of aerosol extinction ($\sigma_a$) integrated over the depth of the atmosphere
$$\tau_a = \int_0^h \sigma_a \, dr = \int_0^h \int_r Q n(r) r^2 \, dr \, dz \qquad (22)$$


where Q is the scattering efficiency and can be obtained from Mie-calculations. The parameter n(r) is the
density of the size distribution and r is the radius of the particle. The vertical integration over height z is over the
depth of the atmosphere (h) and yields
$$\tau_a \sim \sigma_{a,mean} H = Q_{mean} N_a H R^2 \qquad (23)$$


Here $N_a$ is the concentration of aerosols, R is the mean size of the aerosol particles and H is a scaling depth
proportional to the depth of the planetary boundary layer. The proportionality factor includes all vertical
variations in aerosol, size distribution and optical properties. Aerosol extinction can be approximated as
(Eltermann, 1970; Kriebel, 1978, Peterson and Fee, 1981; Wang et al., 2009)
$$\sigma_{a,mean} = \frac{-\log_e(0.05)}{Visibility} \qquad (24)$$


Visibility is a measurable quantity and it provides a means to compute aerosol optical thickness at hourly
intervals from standard weather station observations. This procedure has been used to obtain decadal time series
of the aerosol optical thickness over the Netherlands and China (Boers et al, 2015; Wu et al., 2014). Here, a
universal climatological value for H = 1000 m is used to match the calculations of radiation. We examined the
European Center for Medium Range Weather Forecast Reanalysis (ERA) data (Dee et al., 2011) for changes in
the planetary boundary layer depth. No indications for changes were found in the course of 50 years.
**2.6 Radiative transfer calculations**
Variations or trends in solar radiation under cloudless conditions are mostly caused by variations in the optical
properties and concentrations of aerosols, the ARI. The principle aim here is to assess whether the variations in




optical properties can explain the observed variations is solar radiation. For this purpose, we used a simple
radiation transfer model based on the delta-Eddington two–stream approach, as added complexity in radiative
transfer models will not increase the confidence in our results.

For model calculations, the parameters affecting the radiation are aerosol optical thickness, single scattering
albedo, asymmetry parameter and Ångstrøm parameter. Of these four parameters the first two are the most
important and only the first one can be obtained from observations. It was attempted to derive the single
scattering albedo and its time variation from the aerosol composition in the Netherlands (Boers et al., 2015) but
its precise quantification remains elusive due to its uncertain dependence on aerosol composition, wavelength,
aerosol hygroscopicity and relative humidity. Thus a constant value of 0.90 was used instead. The results of
Boers et al. (2015) indicate that a considerable portion of the reduction in aerosol optical thickness or potential
solar brightening can be attributed to the reduction of sulphate aerosols after the 1980's. Even though the nitrate
values did increase over the same time, their increases cannot completely counterbalance the decreasing
sulphate concentrations.
**2.7 Solar radiation and aerosol-cloud interaction**
Variations or trends in solar radiation emanating from the action of clouds are mostly caused by variations in the
cloud fractional coverage and by variations in the optical properties and concentrations of droplets or ice. The
two main hypotheses for ACI to operate on cloud properties are formulated below as Hypothesis 1 and 2, in the
remainder of this paper referred to as ACI-I, and ACI-II, respectively. ACI-I suggests that variations in cloud
optical properties are attributable to variations in aerosol concentration itself. A massive amount of literature has
been devoted to this subject, but Twomey (1977) is the first one to describe this effect. It is based on a causal
link between changes in aerosol concentration ($N_a$) and cloud droplet concentration ($N_c$). These two parameters
are not necessarily linearly linked: as the amount of aerosol particles increases, it becomes more and more
difficult to raise the supersaturation necessary to activate additional particles. Therefore, $N_c$ and $N_a$ are often
related by means of a logarithmic function or a power law with exponent smaller than one (Jones et al., 1994;
Gultepe and Isaac, 1995), e.g..
$$N_c \sim N_a^{0.26} \qquad (25)$$
Only a limited amount of aerosol particles will be activated to cloud droplets and incipient water droplets all
compete for the same amount of water vapor as they grow. This means that the mean size of cloud droplets
decreases as the number of cloud droplets increases. The consequence for the cloud optical thickness is that:
$$\tau_{c,ACI} \sim H_c N_c^{1/3} \qquad (26)$$

Here $H_c$ is the depth of the cloud and $\tau_{c,ACI}$ is the cloud optical thickness attributable to the aerosol aerosol-cloud
interaction (ACI-I). Thus, compared to Eq.(23) where the equivalent link between aerosol optical thickness and
aerosol number concentration is described the dependence of optical depth to number concentration is much
weaker.

Combining Eqs. (25) and (26) with Eq. (23) we find:



$\tau_{c,ACI} \sim \tau_a^{0.26/3}$ (27)

As the cloud optical thickness $\tau_c$ (which is due to the ACI –I and other causes) can be obtained from inverting
the cloud-base radiative fluxes obtained from Eq. (13), and $\tau_a$ can be obtained from Eqs 23, (24), the validity of
the Eq. (27) can be studied.

ACI-II suggests that increasing $N_c$ will result in suppression of precipitation so that cloud life time and cloud
fraction is increased (Albrecht, 1989). In our analysis, cloud fractional coverage at specific cloud cover is
obtained in a straightforward manner by conditional sampling and counting procedures using hourly cloud data
so that the hypothesis that changes in aerosol results in changes in cloud cover can be tested.
**3 Data analysis**
**3.1 Data sources**
We used quality controlled time series of hourly data of surface radiation, cloudiness and visibility which are
standard output commonly available to the general public and submitted to the traditional climate data
repositories. The surface radiation data consist of 10 second data for shortwave radiation instruments integrated
over the hour. To be consistent with most publications on the subject of trends in radiation, the hourly average is
taken and expressed in $Wm^{-2}$. The visibility is recorded at the end of each hour, either by the Human Observer
(until 2002) or taken from a Present Weather Sensor (PWS, after 2002). Cloud cover is observed by the Human
Observer until 2002 and represents the last 10 minutes of every hour. After 2002 it is observed by a vertically
pointing ceilometer and represents the average of the last 30 minutes of the hour.

A serious concern is that conditional sampling was done on the radiation data in a situation where the
observation that represents the condition (namely whether or not clouds are present), was not taken in exactly
the same time interval as the observation (radiation) itself. Therefore the conditionally sampled data are an
imperfect representation of the true situation.  This is particularly true for rapidly changing cloudiness
conditions. This issue cannot be rectified. However, in this paper exclusive use is made of yearly averages of
conditionally sampled radiation data. For these data, the averaging procedure cancels out data with too much or
too few clouds within the hour of the selected radiation data, so that the variability observed in the data will be
simply enhanced random noise.
**3.2 Metadata**
Table 1 presents the basic metadata of the five principal climate stations in the Netherlands together with the
dates when the collection of radiation data started. The station metadata archive was analyzed from which it was
apparent that initially the regular maintenance and understanding of instruments was inadequate. Typical
problems that needed to be overcome were the build-up of moisture between the concentric glass half-domes,
the removal of dust and bird droppings, the horizontal alignment of the instrument and the proper positioning of
instruments with respect to shading obstacles such as (growing) trees.



Apart from these issues, insufficient (re)calibration of the instruments, irregular replacement / rotation of
instruments from the instrument pool are the reason that the initial years of observation often yielded data of
dubious quality. In the end it was decided to discard all data from the climate stations before the year 1966. The
data from the station De Bilt are of acceptable quality from 1961 onwards, in particular since from that year
onward radiation was measured by two radiometers that were placed side-by-side. However these earlier data
will not be used here because this would induce unacceptable weighting on this station of the radiation average
in the five year period prior to the year 1966.
**3.3 Homogeneity test**
Even though some investigators have attempted with some success to homogenize and gap-fill their data
(Manara et al, 2016) for a small region of the Netherlands with few stations (in our case 5) such a
homogenization procedure is unlikely to be successful. The reason is that it carries the risk of replacing real data
with bogus data which would weigh heavily on the few data time series available. Nevertheless it is instructive
to apply a homogeneity test to understand differences between the time series.
The five radiation time series were analyzed for statistical homogeneity using the Standard Normal
Homogeneity Test (SNHT; Alexanderson, 1986). Instead of applying SNHT directly to each station series, we
used relative testing. Relative testing removes the natural variation from a time series (while assuming that
natural variation is about the same for all locations), which increases the probability of detecting statistically
significant breaks. The SNHT was applied to each station series, reduced with (a) the mean of the four other
station series, and (b) the other four station series separately. The latter would reveal a break in the series. Note
however that the results yield potential statistical breaks, not real ones.
The homogeneity testing was applied to the 1966-2015 period. The results indicate that De Bilt data are
different from the others in the 1966-1975 period, though a possible inhomogeneity reveals itself only in two of
the four relative series. From the metadata there is, however, no reason to doubt the quality of the series of De
Bilt in this particular period. In fact of all five stations the instruments at the De Bilt observatory were probably
maintained in the most optimum way. Also, the series of Eelde appears to be high relative to the other four
station for the 1966 -1972 period although again from the metadata there is no reason to judge the series of
Eelde in this particular period as suspect. Eelde is the most north-easterly station in the Netherlands and data
from this station were compared to the most nearby German station with a long radiation time series
(Norderney, 1967 – 2015). This comparison indicated that Eelde is homogeneous with Norderney, strongly
suggesting that the relative high values of radiation at Eelde in the period 1966 – 1972 are indicative of real
atmospheric variability rather than instrumental problems.
A similar homogeneity test was applied to the standard aerosol optical thickness output from the stations based
on Eq. (23) and (24) which in turn are based on the visibility observations. From these tests it emerges that the
stations Vlissingen and De Bilt depart the most from the average. Furthermore, when all stations are compared,
De Bilt departs the most from the other four. Again these differences can very well imply real differences



between station, such as for example may be the result of local differences in air pollution that influence
visibility (and thus optical thickness).
For the remainder of the research we decided to use the mean of all five stations for the 1966-2015 period. We
studied the sensitivity of the results to leaving out stations and found that even though some details were
different, it did not significantly alter any of the findings and conclusions.
**3.4 Okta and cloud amount**
Even though cloud amount is commonly indicated with the parameter okta, its translation to actual cloud
amount as a fraction is necessary for usage in this paper. According to World Meteorological Organization
guidelines (WMO, 2008) actual cloud amount should be indicated as one okta in case a single cloud is present in
an otherwise completely clear-sky. Similarly, if a single hole exist in an otherwise overcast sky cloud amount
should be indicated as seven out of eight. Therefore, a cloud amount of one okta corresponds to a lower cloud
amount than expected based on the numerical value of one-eighth. Similarly, a cloud amount of seven-eighth
corresponds to a larger value than indicated by its numerical value. Boers et al. (2010) evaluated observed cloud
amounts expressed in oktas with fractional cloud amounts derived from all-sky observation of clouds using a
Total Sky Imager (an instrument sensitive to radiation in the visible part of the solar spectrum) and using a
Nubiscope (an all-sky scanning infrared radiometer). We adhere to the results of their study (their section 2.3,
table 1) where for okta 0-8 the following cloud amounts are given (in percentage): 0.00, 6.15, 24.94, 37.51,
452 50.03, 62.56, 75.18, 95.07, 100.
In the analysis presented in the next section a practical problem occurred in distinguishing between radiation
emanating from a completely clear-sky or from a sky with a single cloud but otherwise clear. In the latter case,
provided that the cloud does not completely block the direct solar beam, it will be impossible to discern whether
the radiative flux would have come from a sky with the okta=0. For this reason it was decided to take data from
c=0 and c=1 together and designated the combined data as 'clear-sky'. A similar argument can be made for the
radiation at the high end of cloudiness. Hence, data from c=7 and c=8 were lumped together as designating an
'overcast' sky.
**3.5 Discontinuity in 2002**
During the year 2002 the Human Observer was replaced by the Present Weather Sensor for visibility
observations and by the ceilometer for cloud observations. While the former transition posed little problems in
the analysis of data, such was not the case for the latter. When observing clouds the Human Observer takes into
account the full 360-degree view of the horizon. A ceilometer only observes a narrow portion of the sky in
vertical direction. Although the half –hour averaging of the cloud observations to some extent compensates for
the absence of instantaneous hemispheric information, the two types of observation represent different methods
of estimating cloud cover so that the conditional sampling of the radiation is significantly affected. For example,
the digital nature of the ceilometer observation results in many more observations in the c = 0 (cloudless) and
the c = 8 (overcast) cloud cover selection bin than obtained from the Human Observer (Boers et al., 2010). As a
result, the selectively sampled radiation data in both okta bins will be contaminated by data recorded under





fractionally cloudy conditions. Contamination by other okta values is also present for data selected for each of
the 1 – 7 okta range but less than for overcast sky conditions.  As a result, the selectively sampled radiation data
showed distinct discontinuities in 2002.

To account for the discontinuity we decided to apply a so-called quantile-quantile correction to the frequency
distribution of cloud coverage from the period after 2002 (during which the ceilometer was operative) and adjust
it to the frequency distribution from the period before 2002 (during which the Human Observer was operative).
The quantile-quantile correction (Li et al., 2010) is commonly used to adjust distributions of meteorological
parameters of numerical models to observed distributions of the same parameters. As a first step cloud cover
data (converted from okta to fractional cloudiness, see section 3.4) from the period 2002 – 2015 was smoothed
by a Gaussian filter with a half-width of two data points (i.e. two hours). This produced a smooth distribution
which, when converted back to okta, yielded a distribution similar but not the same to the okta distribution of
the Human Observer. The next step was to do a quantile-quantile correction on the smoothed data. The
credibility of a quantile-quantile correction depends on whether it can be assured that the average distribution
function as observed by the Human Observer does not change over the break (in case the Human Observer
would have made the observations after the break). Although there were some long-term changes in the
distribution function before the year 2002 they were small enough to assume the invariance of the distribution
function over the break. With the application of the quantile-quantile correction the okta values and hence the
fractional cloudiness values after the break assume new / corrected values that are applied as new / corrected
discriminators in the selection of the radiative flux.

As a proof of soundness of the procedure we applied the quantile – quantile correction and recomputed the
fractional cloudiness as the summation $\sum_1^8 f_i c_i = \bar{c}$ (see discussion beneath Eq. (6)) and compared the result to
satellite observations derived from successive NOAA-satellites (Karlsson et al, 2017). Figure 1 shows the
results.

The NOAA data (red line) comprises an average over the Netherlands and have been bias-corrected. It is clear
that the surface data (black line) which are break-corrected after the year 2002 provides an excellent agreement
to the NOAA data when compared to the data which are not-break corrected (blue line).  Note also that the data
that are not break-corrected show a downward trend in cloudiness while the break-corrected data show an
upward trend. These results are thus at odds with observations in Germany close to the Netherlands (Ruckstuhl
et al, 2010)  where cloud cover seems to be declining at least until 2010.
**4 Results**
**4.1 Decomposing the all-sky radiative fluxes**
As a first step in understanding the relative impact of clear and cloudy skies on the all-sky radiative flux it is
instructive to examine the manner in which the top-of-atmosphere (TOA) radiative flux is reduced by the
various constituents and scattering and absorption mechanisms in the atmosphere (Figure 2). The combined



effect of all these processes is responsible for reducing the TOA radiative fluxes down to the observed all-sky
radiative flux as indicated by the white line at the bottom of the figure. Figure 2 is a combination of calculations
and observations. Observed are the all-sky flux ( the white line at the bottom of the Figure) and the clear-sky
flux (the white line in the middle). Starting from the top downward, the first reduction of the TOA flux is due to
Rayleigh scattering, namely downwards from 274 to 253 $Wm^{-2}$. Continuing downwards ozone absorption is
responsible for a further reduction from 253 to 246 $Wm^{-2}$. Next water vapor absorption reduces the radiative
flux by a further 39 $Wm^{-2}$ from 246 to 207 $Wm^{-2}$. These three decrements were calculated from inputs from
ERA (for the ozone and water vapor absorption) or surface pressure observations (for the Rayleigh scattering).
The next reduction is due to the aerosol scattering and absorption which takes the radiative flux further down to
the observed clear-sky flux (or more precisely the proxy) from 207 $Wm^{-2}$ to ~170 $Wm^{-2}$ around 1970 or to ~185
$Wm^{-2}$ near 2015 with a steady increasing value during the intermediate years. The solid white line in the middle
of the plot represents the clear-sky flux. The rest of the reduction from the clear-sky radiative flux to the all-sky
flux is entirely due to the action of clouds. The observed clear-sky (proxy) shortwave radiation shows that about
13.6 $W m^{-2}$ has been added to the clear-sky radiation over a period of 5 decades. A trend value at 2.78±0.50 W
$m^{-2}$ / decade was calculated by the Mann-Kendall test (Kendall, 1975) after the time series was first
decorrelated. The uncertainty value attached to the trend is a test of significance indicating the 95% confidence
interval of the calculated slope line. The upward trend in clear-sky radiation is thus deemed to be strongly
significant. The lower white solid line represents the all-sky radiation which is derived straight from the publicly
available climate data sources. It shows considerable short-term variations but overall there is a positive trend.
The trend value was calculated as 1.81±1.07 $W m^{-2}$ / decade and is thus also considered significant.
When comparing the different contributions there are three important points to be considered. First, the
combined effects of Rayleigh scattering, ozone and water vapor absorption is constant over time. Even though
there is a slight increase in water vapor path over the 50-year period, this is not reflected in any discernable
decrease in radiative flux. Second, despite the absence of any significant trends in the respective radiative
reductions they make up a very substantial part of the overall reduction from the TOA radiative flux to the all-
sky flux (40 – 50%). Third, the two-pronged action of clouds by 1) blocking part of clear-sky flux in reaching
the surface and b) by scattering radiation inside the clouds is considerably larger than the action of scattering
and absorption of radiation by aerosols in reducing the TOA radiative flux. The former ranging from double the
latter at the beginning of the period to triple the latter at the end of the period.
Figure 3 shows the measured all-sky radiation and the proxy clear-sky and weighted cloud-base radiation.
Linear regression lines (blue) as well as a 21-point Gaussian fit (red) are shown in the figure. There is a weak
minimum in all-sky radiation at 1984 which is matched by a minimum in cloud-base radiation near 1982 – 1984.
In contrast the clear-sky radiation has an upward trend throughout the entire period. All trend are significant
when taken over the entire period.
Figure 4 shows the key result of this paper namely the reconstruction of the trend in the all-sky (proxy) flux out
of its three main components as formulated in Eq. (21). Here, the last term, a cross correlation term is not shown





on account of its very small yearly values (less than 0.5 W m$^{-2}$). The black curve shows the variation in all-sky
proxy radiation as a function of time. Note again that this function is slightly different from the real all-sky
radiation data as its construction is based on the proxy data. Even so, the fluctuations and trends in the proxy
data are clearly very close to the fluctuations and trends as observed in the real all-sky data of Figure 3.
However, the Gaussian-filtered data indicates that the weak minimum in the original data is replaced by a (close
to) constant value in the proxy data. The red curve is the contribution to the trend in all-sky proxy radiation due
to the trend in cloud amount. Cloud amount is increasing and as a consequence the overall trend is negative. The
green line is the contribution to the trend in all-sky proxy radiation as a result of the positive trend in clear-sky
proxy radiation, but modulated by the average fraction of time that it is actually clear (32%). The blue line is the
contribution to the trend in all-sky radiation as a result of the positive trend in proxy cloud-base radiation. It has
a broad minimum, but modulated by the fraction that it is cloudy on average (68%). Each curve represents a
perturbation with respect to its average and the tick marks represent intervals of 10 W m$^{-2}$.

A number of intermediate conclusions can be drawn at this point:
1. The cloud-base and cloud cover contributing trends are of the same order of magnitude whereas the
clear-sky trend contribution is less significant than either one of them.
2. As the mean fractional cloudiness at 0.68 is larger than 0.50, the contribution to the all-sky flux due to
a trend in cloud-base radiation has a comparatively larger weight than the contribution of the trend in
clear-sky radiation.
3. The increase in cloud cover results in a negative trend contribution to the trend in all-sky (proxy)
radiation which thus dampens the strong trend contribution due to the increasing cloud-base proxy
radiation.
4. The short-term variations in all-sky radiation are almost entirely due to the short-term variations in
fractional cloudiness.
5. The weak minimum (constant) in all-sky (proxy) radiation is strongly linked to trends in clouds, but not
as much to the trend in clear-sky radiation.

Table 2 summarizes the results of the trend analysis. Here, also a subselection is made according to the time
period over which trend analysis is performed. Significance is indicated in the last column.

Inspection of the table indicates that none of the trends (including those of the clear-sky proxy radiation) is
significant in the period 1966 – 1984. All significant trends occur in the period 1984 – 2015. Two-thirds of the
strong upward trend in cloud-base proxy radiation is offset by the cloud fraction term in the same period.
To our knowledge these calculations are the first of their kind and demonstrate the relative importance of the
impacts of clear and cloudy skies on the all-sky radiation. Trend values for the all-sky radiation all fall within
the bounds of Lorenzo-Sanchez et al. (2015) given by their comprehensive summary of Europe's observations.
For the clear-sky radiation the trend is positive throughout the entire period and the absence of a curvature
matching that of the all-sky radiation does not suggest a very strong causal link with it. In contrast the curvature
of the cloud-base radiation curve much more resembles that of the all-sky radiation. Because the fractional cloud





cover term partly compensates the strong upward trend of the cloud-base curve after 1985, it strongly suggests
that for the Netherlands cloud processes are the dominant factor that impact the shape of the all-sky time series.

**4.2 Aerosol-radiation interaction (ARI)**

To investigate the possibility of aerosol-radiation interaction the median aerosol optical thickness is derived
from the visibility observations. Next radiative transfer model calculations were performed to compute the solar
radiation. Figure 5 shows the time series of median aerosol optical thickness for the Netherlands. To about 1985
the optical thickness has a weakly downward trend albeit that there are considerable year-to-year variations.
After 1985 there is a distinct downward trend that remains present until the end of the time series in 2015.
Overall trend is -0.032 per decade and is significant.

Figure 6 shows the results from radiative transfer computation compared to the clear-sky flux. The solid black
and accompanying shading represents the best fit through the data (the points connected by a black line). The
blue line is the result of calculating the clear-sky radiation using the aerosol optical thickness in Figure 5 as an
input, with a fixed value of the single scattering albedo of 0.90. The calculations indicate a remarkable
agreement with the observed clear-sky radiation. The blue line falls entirely within the shaded area of
uncertainty of the slope through the data.

The accuracy of the modeled radiation curves is dependent upon the accuracy of the optical thickness derived
from the visibility observations and the value of the single scattering albedo. If the scaling depth used to match
the optical thickness observations to satellite and surface-base radiation data (Boers et al., 2015) is changed, so
will the position of the model output (blue line) change with respect to the clear – air data ($\delta SW = 5 - 6$ W m$^{-2}$
for $\delta\tau = -0.1$).

There is however no useful information on the time-dependence of the single scattering albedo, the mean value
of which is not clear either. The value of 0.90 as used here reflects a compromise between the necessity of
having to assign it a value less than one due to the presence of radiation absorbing aerosols (Black Carbon and
Organic Aerosols), and the prevalence of pure scattering aerosols in an environment of high relative humidity
(sulfates and nitrates) which tend to keep the single scattering albedo at a high value.

However, the overall conclusion is that the reduction in aerosol concentration resulting in a reduction in aerosol
optical thickness is a very strong candidate cause explaining the overall increase in clear-sky solar radiation.
This implies that there is a compelling argument that ARI i.e. the direct aerosol effect is responsible for the
decadal change in clear-sky radiation.

**4.3 Aerosol-cloud interaction (ACI)**

Concerning ACI-I we plotted the left and right sides of the function described in Eq. (27). Here (Figure 7) the
cloud optical thickness for clouds has been derived from the monotonic relationship between solar radiation and
cloud optical thickness and using the mean weighted cloud-base radiation (bottom curve in Figure 2) as the
radiative input. The cloud optical thickness that is thus derived constitutes the left side of Eq. (27). The right



side of Eq. (27) is based on the aerosol optical thickness data as shown in Figure 5. According to Figure 7, there
is indeed an indication that there may be a link between the two optical thicknesses but the regression line has a
larger slope than suggested by Eq. (27). This suggests that there may be other mechanisms that play a role in
changing the cloud optical thickness. The most likely candidate responsible for these additional changes is a
decadal thinning of clouds. However, there is no confirmation by independent data sources suggesting that such
thinning has indeed taken place over the course of five decades.

Under ACI-II cloud amount is governed by precipitation. Here a reduction in aerosols over time would increase
the size of cloud droplets, thus enhancing the fall-out of liquid water and thus reducing cloud amount. However,
data shown in Figure 1 indicate that cloud fraction is increasing after 1985 when at the same time the aerosol
optical thickness decreases. This does not necessarily mean that ACI-II is not operative, but that other factors
(such as large scale synoptic changes) at least overwhelm any possible cloud cover changes due to ACI-II.

## 639    5 Discussion and conclusions

Our derivation of a trend equation for the all-sky radiation shows that there are five parameters that influence
the trend, namely 1) a trend in fractional cloudiness, 2) a trend in clear-sky radiation, 3) a trend in cloud-base
radiation, 4) the decadal mean of the fractional cloudiness, and 5) the difference between the decadal means of
the cloud-base and the clear-sky radiation. It is therefore not surprising that it has been difficult up to now to
come up with any firm conclusions about the relative importance of trends in clouds or clear-sky radiation in
contributing to the trend in all-sky radiation. This situation is further hampered by the derivation of clear-sky
and cloud-base radiation, requiring a specialized analysis removing the year-to-year internal fluctuations in
radiation estimates. These are the results of periodic synoptic conditions that favor certain cloudiness conditions.
An analysis of annual means of radiation selected under specific okta values will produce unrealistic results, as
noted by Ruckstuhl et al (2010). In order to overcome this last issue we have cast the problem of estimating
annual mean radiation in a two-dimensional framework with cloud fraction (okta) and cosine of solar zenith
angle as the two controlling variables. A proxy radiation is derived by fitting per okta value a function that is
solely dependent upon cosine of zenith angle. Next annual means are computed using the annually constant
distribution of cosine values. Stable values of radiation ensue from which trends can be calculated.

Our analysis comprises 50 years of hourly radiation, cloudiness and visibility data at the five principal climate
stations in the Netherlands. We summarize the main conclusions of this work.

1)    The three most important mechanisms reducing the top-of-the-atmosphere radiation to the observed all-

sky radiation are absorption of radiation by water vapor, and scattering and absorption by aerosols and

clouds. Over the Netherlands the reduction in radiation due to water vapor absorption is actually larger

than from aerosol scattering and absorption. However, as there is no trend in water vapor, there is no

trend in the all-sky radiation due to trends in water vapor.

2)    Trends in clear-sky, cloud-base radiation and fractional cloudiness are all important in contributing to

the trend in all-sky radiation.



3) Over the Netherlands the clear-sky trend is weighted by 0.32 which is one minus the decadal mean fractional cloud cover and the cloudy-sky trend is weighted by 0.68 (i.e. the decadal mean of fractional cloudiness). Therefore, in the Netherlands a trend in cloud-base radiation has double the weight of a clear-sky radiation trend in contributing to the all-sky radiation trend. Thus, in a general sense this means that the actual value of fractional cloudiness, which has a strong regional dependence, exerts a considerable control over the relative importance of clear-sky and cloud-base radiation trends.

4) Over the Netherlands the trend in fractional cloudiness is significantly positive in the period after 1985 and because this trend is multiplied by the (negative) difference between the decadal means of cloud-base and clear-sky radiation, it contributes as a negative trend to the trend in all-sky radiation. As the literature suggests (f.e. Norris, 2005) there are significant regional differences in long term trends in cloud cover, so it indicates that strong regional differences will exist in its contribution to the trend in all-sky radiation.

5) As found in most studies (see summary of Lorenzo-Sanchez et al., 2015), a minimum in all-sky radiation is found around 1985. The negative trend of -1.4 $\mathrm{W\,m^{-2}}$ up to 1985 is weaker than the average of Europe (-2.5 $\mathrm{W\,m^{-2}}$). The upward trend from 1985 onwards of 2.3 $\mathrm{W\,m^{-2}}$ is also weaker than the average of Europe (3.2 $\mathrm{W\,m^{-2}}$).

6) The minimum in all-sky radiation is not matched by a corresponding minimum in clear-sky proxy radiation. An increasing trend of 1.22 $\mathrm{W\,m^{-2}}$ is found over the earlier period which increased to 3.40 $\mathrm{W\,m^{-2}}$ later on. After significant amounts of local natural gas were found in the late 1950s the Netherlands were a very early (1960 – 1965) adapter to cleaner fuels which may explain the increase in clear-sky radiation in the earlier period (1966-1985).

7) The trend in cloud-base radiation has a similar shape as that of the all-sky radiation. It is weakly negative before 1985 (-0.77 $\mathrm{W\,m^{-2}}$) and strongly positive thereafter (4.94 $\mathrm{W\,m^{-2}}$). Consequently, the conclusion is justified that the curvature /weak minimum in all-sky radiation around 1985 is caused mostly by the cloud-base radiation.

8) As our techniques are able to isolate the clear-sky radiative component it has been possible to study the attribution of changes in aerosol content to the observed trend in clear-sky radiation. Radiative transfer calculations demonstrate that the increase in clear-sky radiation can be completely explained by a concomitant decrease in aerosol optical thickness. This strongly suggests that the ARI (the direct aerosol effect) is a prime candidate to explain the observed increase in clear-sky radiation.

9) Similarly, ACI-I and ACI-II have been studied to understand their potential impact on the all-sky radiation. Neither is shown to have a dominant contribution to the trend in the overall all-sky flux but the potential influence of ACI-I and ACI-II cannot be ruled out by the data: There may be other influencing mechanisms that mask the impact of ACI-I and ACI-II such as decadal changes in cloud thickness and fractional cloudiness as a result of large-scale synoptic phenomena.

Prerequisite for our method to work is the availability of simultaneous time series of radiation, cloudiness and visibility. The first two are necessary to resolve the difference between clear and cloudy-sky signals in the radiation data, a method which in this paper has been called the determination of 'proxies'. Additional observations of visibility are necessary to understand the possible influence of aerosols on radiation.





There are a number of ways to improve and/or facilitate this work in the future:

1)   The practice of observing different parameters simultaneously can be improved by a more optimum
consideration of the impact of one parameter on another. For example aerosols and clouds impact
radiation, but radiation is recorded as an hourly average, while clouds and visibility parameters are
recorded as averages of smaller time intervals. Often these different recording and averaging intervals
are based on WMO standards. Yet, they inhibit the analysis and interpretation of their physical links. It
would be better if averaging times were standardized more uniformly or if the basic data underlying the
averages become available.

2)   The relative contribution to the all-sky radiation of cloud thickness remains unclear. Therefore, the
potential impact of ACI-I and ACI-II cannot be unambiguously quantified. The best way to resolve this
issue is by adding observations of clouds using a cloud radar and a cloud lidar. As clouds are largely
transparent to radar probing cloud thickness and its long-term variations can thus be derived. Here,
super-sites such as those of the Atmospheric Radiation Measurement program and CloudNet, or long-
term data from CloudSat could be of great assistance. Passive radiation data from satellites are less
suitable as they only record radiation emanating from the top of clouds or from the layer just beneath
cloud top.

3)   The impact of changes in the single scattering albedo is unclear. This situation is best resolved by
direct observations of the single scattering albedo including its wavelength dependence. However, this
suggestion only works for future studies as observations of single scattering albedo have hardly been
performed in the past. It may be that regional modelling of past aerosol composition and physical and
optical properties may alleviate the historical lack of single scattering albedo data.



**Data availability**

The data used in this paper can be downloaded from the KNMI website:

http://www.knmi.nl/nederland-nu/klimatologie/uurgegevens

**Acknowledgments**

We acknowledge the use of EUMETSAT's CMSAF cloud climatology data sets. We much appreciated discussions with Jan Fokke Meirink who made us aware of this data set and who instructed us on its use in this analysis. We also appreciated discussion with Wiel Wauben about the break analysis.





**Tables**

**Table 1**: **Details of the stations and the introduction data of the radiometers.**

| Station | WMO nr. | LAT (N) | LON (E) | ALT (m) | Introduction date |
|---|---|---|---|---|---|
| De Kooy | 06235 | 52.924 | 4.785 | 0.5 | 24 September 1964 |
| De Bilt | 06260 | 52.101 | 5.177 | 2.0 | 10 May 1957 |
| Eelde | 06280 | 53.125 | 6.586 | 3.5 | 2 October 1964 |
| Vlissingen | 06310 | 51.442 | 3.596 | 8.0 | 10 April 1962 |
| Maastricht | 06380 | 50.910 | 5.768 | 114.0 | 5 March 1963 |

739
740





**Table 2. Summary of trend analysis. Except for the fractional cloudiness, all parameters have W m$^{-2}$ / decade as a unit. Whether or not the indicated trend is significant is indicated by the star in the column 'uncertainty'.**

| Type | Period | Trend | Uncertainty |
|---|---|---|---|
| Fractional cloudiness | 1966-2015 | 0.0097 | 0.0062* |
|  | 1966-1984 | -0.0055 | 0.0344 |
|  | 1984-2015 | 0.0205 | 0.0117* |
| All-sky radiation | 1966-2015 | 1.81 | 1.07* |
|  | 1966-1984 | -1.40 | 4.19 |
|  | 1984-2015 | 3.30 | 1.55* |
| All-sky proxy radiation | 1966- 2015 | 1.89 | 0.78* |
|  | 1966- 1984 | 0.39 | 3.86 |
|  | 1984- 2015 | 2.30 | 1.68* |
| Clear-sky proxy radiation | 1966-2015 | 2.78 | 0.50* |
|  | 1966-1984 | 1.22 | 2.14 |
|  | 1984-2015 | 3.46 | 1.35* |
| Cloud-base proxy radiation | 1966-2015 | 3.43 | 1.17* |
|  | 1966-1984 | -0.77 | 2.01 |
|  | 1984-2015 | 4.94 | 2.30* |
| Fractional cloudiness term | 1966-2015 | -1.06 | 0.67* |
|  | 1966-1984 | 0.43 | 3.30 |
|  | 1984-2015 | -2.22 | 1.19* |
| Clear-sky proxy term | 1966-2015 | 0.88 | 0.16* |
|  | 1966-1984 | 0.39 | 0.68 |
|  | 1984-2015 | 1.09 | 0.43* |
| Cloud-base proxy term | 1966-2015 | 2.35 | 0.80* |
|  | 1966-1984 | -0.53 | 1.38 |
|  | 1984-2015 | 3.37 | 1.57* |



**Figures**

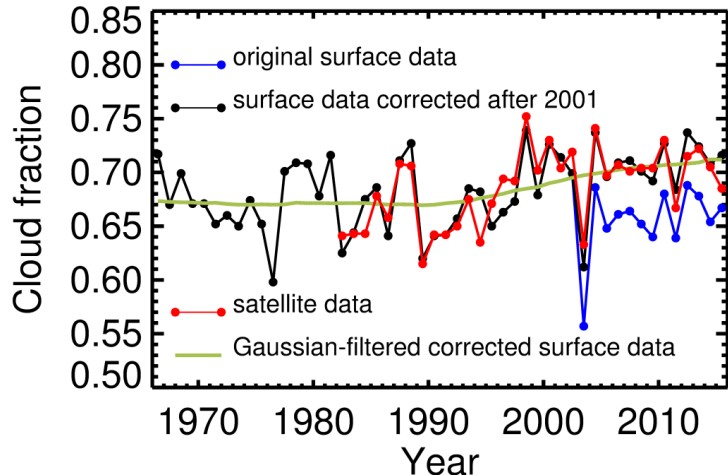


**Figure 1**. **Surface-based cloud fraction estimates versus satellite-based estimates.**





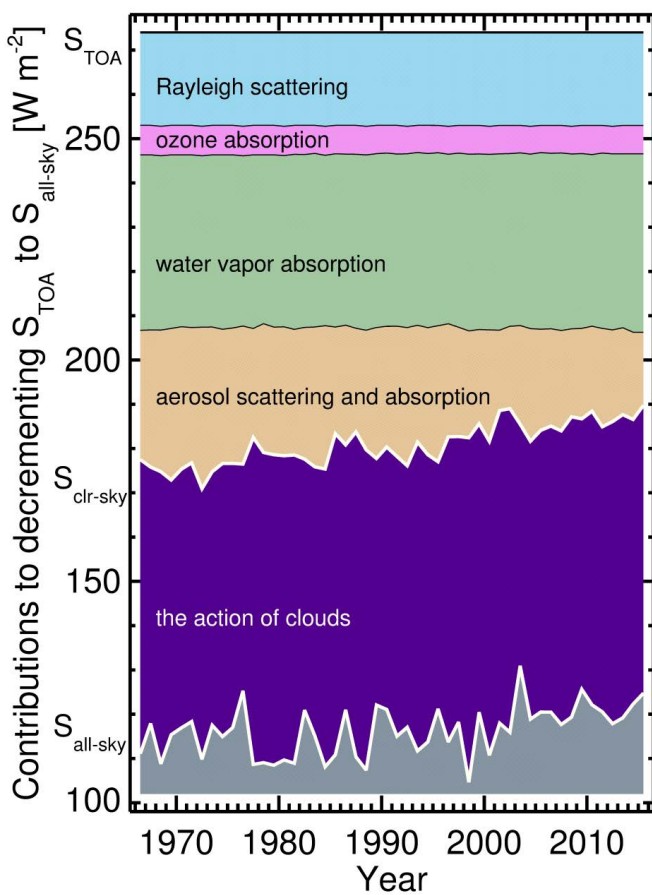

**Figure 2. Impact on all-sky flux due to Rayleigh scattering, ozone absorption, water vapor absorption, aerosol scattering and absorption and the action of clouds.**









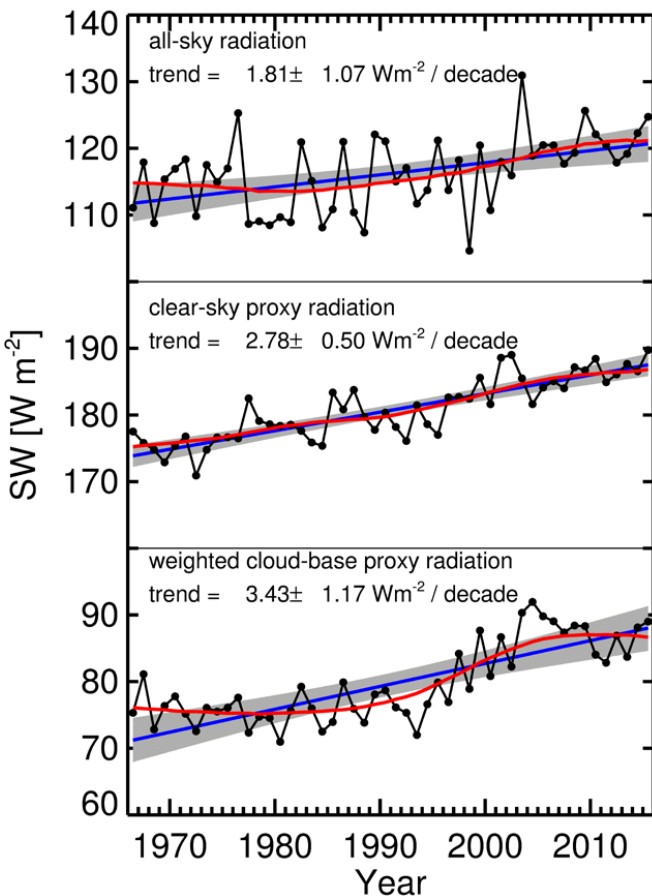


**Figure 3. All-sky, clear-sky proxy and cloud-base proxy radiation as a function of time. Blue lines are the**

**regression fits with the grey area as the uncertainty around the fit. The red lines are 21-point Gaussian**
**filter smoothers.**




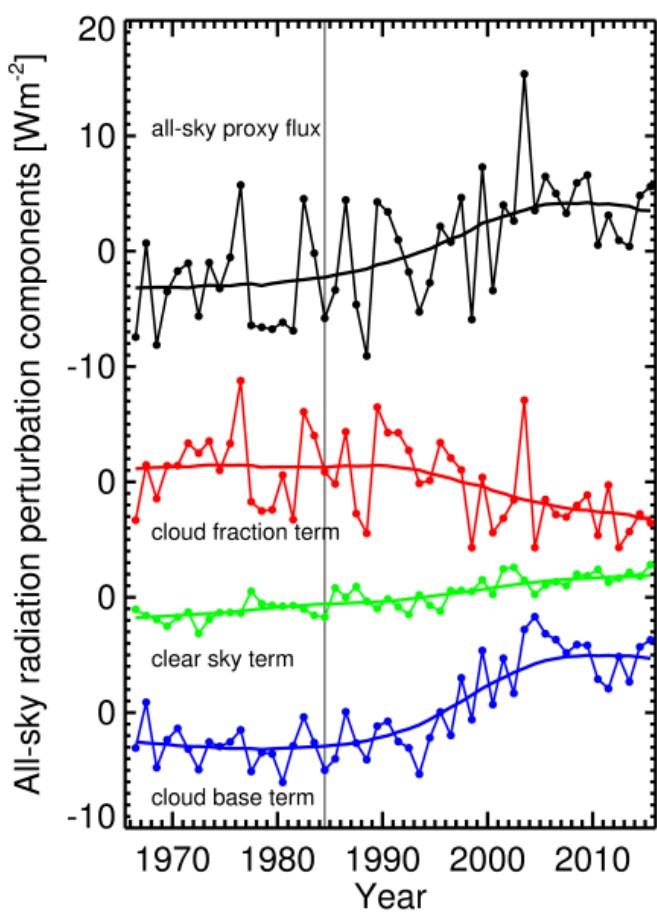


**Figure 4. All-sky radiation perturbation components. Terms are indicated in the graph. 21-point**

**Gaussian filter smoothers are drawn through the curves.**






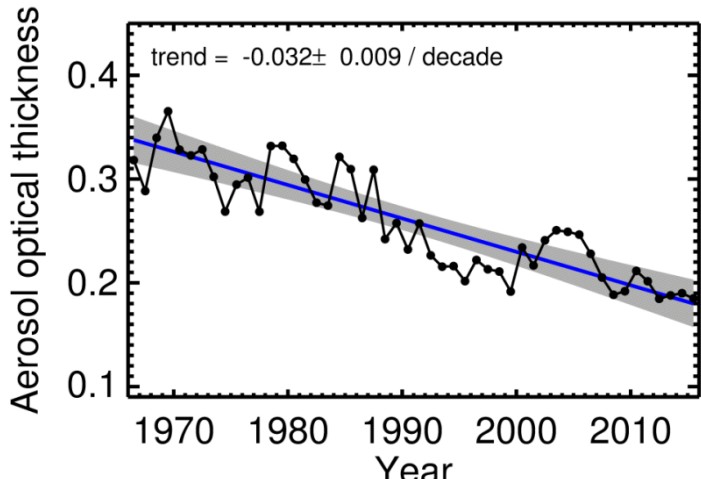


**Figure 5**. **Aerosol optical thickness derived from visibility observations.**

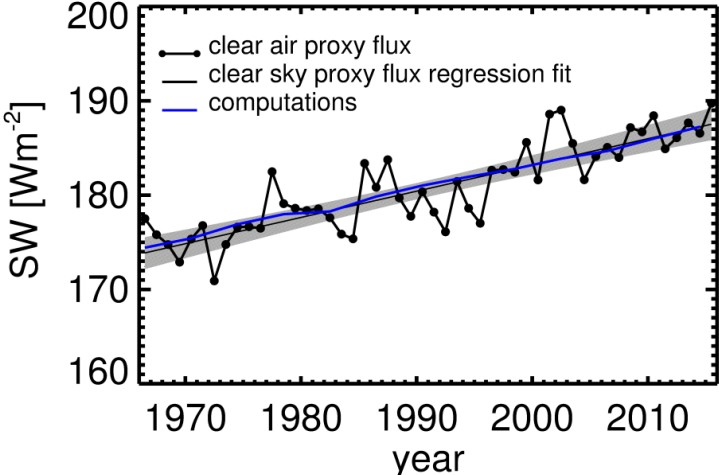


**Figure 6**. **Clear-sky radiation observations matched by radiative transfer computations.**





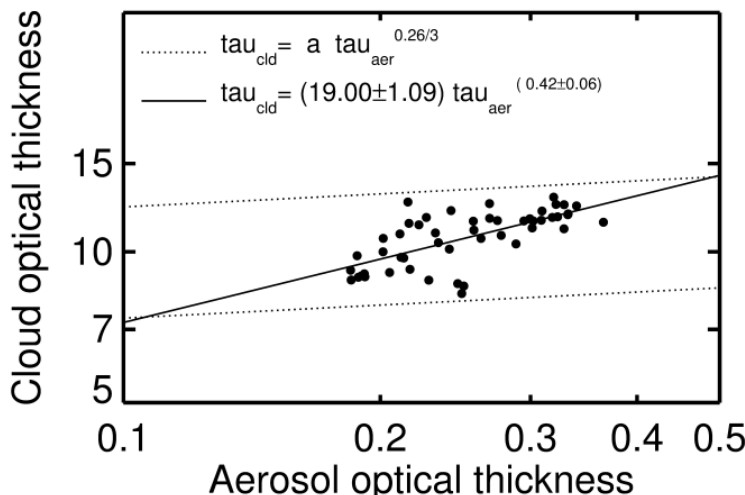


**Figure 7. Cloud optical thickness as a function of aerosol optical thickness. The broken lines are the**

**suggested dependencies of the two optical thicknesses assuming that ACI-I is valid. The solid line is the**

**actual fit through the data.**




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
