# Peer review of "Impact of aerosols and clouds on decadal trends in all-sky solar radiation over the Netherlands (1966 – 2015)"

_Atmospheric Chemistry and Physics, 2017_

## Referee Comment (RC1) · Anonymous Referee #1 · 7 Mar 2017

Review of the manuscript: "Impact of aerosols and clouds on decadal trends in all-sky solar radiation over the Netherland (1966-2015)" by Boers et al. The authors made a good work in analyzing 50-year hourly dataset of global radiation, cloudiness and visibility over the Netherland in order to quantify the contribution of aerosols and clouds to trends in all-sky radiation. They show that all trends in fractional cloudiness, clear-sky and cloud-base radiation contribute significantly to the observed trend in all-sky radiation. I suggest to consider this paper for publication after the following issues are addressed: Specific comments: - Le length of the manuscript could be reduced (especially sections where the methods are described). In this way it will be easier to read the paper and to follow the discussion. - Line 121: the authors write that all-sky

radiation is a function of three components: clear-sky radiation, cloud-base radiation and fractional cloudiness. How do you think that the results could change considering also the type of clouds and not only their extent? - Line 262: the right hand side of the equation has four components. Only three of them are discussed (lines: 264-269). At line 269, the authors write that the fourth term is not shown. Clarify this point. - Line 325: How are estimated the last two parameters used for model calculations? - Line 430: How does the present weather sensor work? Why does the change from human observations to automatic sensor introduce a break in cloudiness series and not in visibility series? - Lines 568-570: How do you explain this result? Technical corrections: - Check the reference at line 77; - Line 105: It is the first time that the abbreviation ACI is used in the text so it is necessary to define it (even if it is already defined in the abstract); - Line 226: Define all the parameters in equation 16; - Some additional references are necessary, for example at lines: 298, 303, 315, 323, 350; - Check the reference at lines 584 and 676.

---

## Referee Comment (RC2) · Anonymous Referee #2 · 17 Mar 2017

Second review of "Impact of aerosols and clouds on decadal trends in all-sky solar radiation over the Netherlands (1966-2015)," by R. Boers, T, Brandsma, and A. P. Siebesma

General comments:

This paper shows an innovative approach to isolate the sources of dimming and brightening for the Netherlands over a 50-year period. It is of high scientific significance because the trends of dimming then brightening over the period of study are well known global phenomena, but their causes are not universally consonant. Problems in documenting dimming, especially, in the early part of the period of study are hampered by a lack of appropriate data that have been generally available during the recent brightening period. In my opinion, the authors have ably used available sources of data to create credible proxies and correct data appropriately (e.g., cloud fraction over the transition from human observers to ceilometers) to study dimming and brightening over the Netherlands and impressively determined the relative contributions of aerosols and clouds to those phenomena. The mathematics used is impressive but cumbersome and could be simplified for the reader by including only the final equations in the main text of the paper, with complete descriptions of their components and the details of the derivations placed in an appendix. That is only a suggestion. In addition, more frequent reminders to the reader of the time periods that terms discussed represent would be useful.

One major concern is that the primary results presented in Table 2, and summarized in the abstract, are not intuitive and require more explanation. That is, the 50-year trends in clear-sky and cloud-base radiation are both greater than the 50-year trend in all-sky radiation. Intuitively, the former two trends should add to the all-sky trend. However, when examining sub-trends during the periods of dimming and brightening separately, which can be done from the results presented in Table 2, the component trends (i.e, clear-sky and cloud-base) do sum better to the reported all-sky trends for those sub-periods, within the margins of error presented. One potential problem I see in your analysis is in your interpretation of eq. 21. As stated on lines 256-257, the over-bars in the equation represent 50-year means and the primed quantities represent yearly deviations from decadal averages. This inconsistency may lead to problems. Another potential problem I see is that the weights applied to the clear-sky term (.32) and the cloud fraction term (.68) represent the fractional periods of clear and cloudy conditions over the entire 50-year period. Since those fractions likely change through the 50-year period, I believe it would be beneficial to analyze eq. 21 over decadal periods, using decadal means, yearly deviations from decadal means, and decadal weights of fractional clear-sky and cloudy periods to compute $S'(y_k)$. Then, the means and deviations used would be internally consistent and the fractional mean periods of clear and cloudy skies would be appropriate to the decade being analyzed.

Specific comments:

Abstract:     As detailed in the general comments, the differences among the three trends listed needs more explanation.

l. 28-30      Brightening in the U.S. from the mid 1990s to ~2011 is attributed primarily to a reduction in cloud cover in Long et al. (2009), Augustine and Dutton (2013), and their results are based on data alone.

                         Why mention that climate models are not capable of reproducing these trends. Are climate models even capable of resolving the cloud physics necessary to resolve the various cloud types and cloud cover responsible for dimming and brightening? I doubt it.

l. 36-42      Wang et al. (2013) has clearly shown that errors associated with single black detector pyranometer measurements adversely affect trends in solar radiation. In this respect, GEBA data are not of unmistakable quality.

l. 73-73      Does your statement that cloud cover data are collected simultaneously with radiation data apply to the Netherlands? That may not be true for most of the radiation stations over the globe.

l. 105      Define acronyms

l. 132-133      How is a representative cosine of the solar zenith angle (SZA) determined for a particular hour? Do you use the cos(SZA) at the midpoint of the hour or do you average the SZAs within the hour period? Averaging SZAs at low sun does not provide a good "representative" SZA for the hour.

l. 199      Change "to suggesting that" to "that suggests"

l. 220-221      To get proxy data for $S_{cj}(y_k)$ I assume that you plot the data corresponding to the variables in eq. 15 in a way analogous to Langley plots, and then use the resulting relationship to generate the proxy data per okta. Correct? If so, I assume that the scatter, and thus uncertainty, of those plots would be small for clear-sky data and larger for oktas 1 through 8. Those uncertainties would define the error in the various terms of eq. 21. Were those uncertainties incorporated into your analysis? They should at least be presented in some form—if possible. If my interpretation of your proxy data generation is wrong, please better explain your method in the paper.

l. 249      This sentence needs to be reworded.

l. 270      The 4[th] term is shown.

l. 321      "is" should be changed to "in"

l. 355      Since you are discussing aerosol optical depth and cloud optical depth in the same sentence, references to each should be specific. In this case, insert "cloud" in front of "optical depth" on this line.

l. 366-368    By "cloud fractional coverage at specific cloud cover," do you mean the transformation of okta observations to fractional cloud cover? If so, please state this more clearly.

l. 414    The sentence beginning with "The SNHT was applied …" is difficult to understand. What do you mean by "reduced with?" and how does that apply to the a) and b) permutations?

l. 516    What is ERA?

l. 537    "b)" should be changed to "2)"

l. 555    The sentence beginning with "Cloud amount is increasing…" is counterintuitive. It would benefit by inserting "in solar radiation at the surface" after "overall trend."

l. 560    The large tick marks represent 10 Wm$^{-2}$.

Wang, K., R. E. Dickinson, Q. Ma, J. A. Augustine, and M. Wild (2013), Measurement methods affect the observed global dimming and brightening, *J. Climate*, 26, 4112-4120, doi:10.1175/JCLI-D-12-00482.1

Augustine, J. A., and E. G. Dutton (2013), Variability of the surface radiation budget over the United States from 1996 through 2011 from high-quality measurements, *J. Geophys. Res.,118,* doi:10.1029/2012JD018551.

Long, C. N., E. G. Dutton, J. A. Augustine, W. Wiscombe, M. Wild, S. A. McFarlane, and C. J. Flynn (2009), Significant decadal brightening of downwelling shortwave in the continental United States, *J. Geophys. Res.,* 114, D00D06, doi:10.1029/2008JD011263.

---

## Author Comment (AC1) · 8 May 2017

**Authors response to Anonymous Referee #1**

**Anonymous Referee #1**

Review of the manuscript: "Impact of aerosols and clouds on decadal trends in all-sky solar radiation over the Netherland (1966-2015)" by Boers et al. The authors made a good work in analyzing 50-year hourly dataset of global radiation, cloudiness and visibility over the Netherland in order to quantify the contribution of aerosols and clouds to trends in all-sky radiation. They show that all trends in fractional cloudiness, clear-sky and cloud-base radiation contribute significantly to the observed trend in all-sky radiation. I suggest to consider this paper for publication after the following issues are addressed:

Authors response: We thank the referee for the comments. Below follow the comments and our answers to them:

Specific comments: - Le length of the manuscript could be reduced (especially sections where the methods are described). In this way it will be easier to read the paper and to follow the discussion. –

Author's response: This point was also brought up by Referee #2. The authors have decided to put the Method sections 2.1, 2.2, 2.3, and 2.4 in an appendix and only write down the end result (including a description of it) of the final equations, namely Eq. (17) and Eq. (21). This is a substantial reduction in the main text which improves the flow of the manuscript.

Line 121: the authors write that all-sky radiation is a function of three components: clear-sky radiation, cloud-base radiation and fractional cloudiness. How do you think that the results could change considering also the type of clouds and not only their extent?

Author's response: Interesting point. If some cloud would have shifted from ice to liquid within those fifty year, the microphysics would have changed [but to an unknown extent]

which in turn could have impacted the radiation at the surface. However, then it needs to be quantified how such changes took place and perhaps more importantly how it would have impacted the radiation. This is not a feasible subtopic within this paper. Nevertheless, we decided to put in a statement alerting the reader to this potential issue.

*Long term changes in cloud type could perhaps affect cloud optical properties (liquid water versus ice water) but their influence on trends is unknown and not studied here.*

Line 262: the right hand side of the equation has four components. Only three of them are discussed (lines: 264-269).

Author's response: This was a remnant of a previous version of the paper. We corrected this as there are indeed four terms. All four are described now.

At line 269, the authors write that the fourth term is not shown. Clarify this point.

Author's response: This point is directly linked to the previous one. The correction to the text was made by removing '*not shown here*' in line 270.

Line 325: How are estimated the last two parameters used for model calculations? -

Author's response: They come from the analysis of the Boers et al., 2015 Environmental Research Letters paper. This is now referenced:

*The asymmetry parameter and the Ångstrøm parameter are set to 0.69 and 1.5 respectively to reflect typical aerosol values derived for the Netherlands (Boers et al., 2015).*

Line 430: How does the present weather sensor work? Why does the change from human observations to automatic sensor introduce a break in cloudiness series and not in visibility series?

Author's response: A statement was put in described the working of this instrument:

*The PWS detects the forward scattering of light emitted by a Near Infrared Light Emitting Diode under an angle of 42º*

We suspect that the simplicity of the PWS instrument in comparison with the ceilometer has much to do with the ease of transition from human observer to instrument. In both cases an overlap period of two years was used to assess their performance. But for the PWS a simple adjustment will probably have sufficed. However, at an early stage it was noted that the transition to ceilometer posed serious problems mostly the result of the fact that the sky coverage of an individual ceilometer observation is a couple of square meters or less, while a human observer covers at least 25 km^2 if not more. How to manage such a transition for selected cloud cover is much more difficult than a simple PWs adjustment.

We decided not to amplify this point further in the text, except by stating for the PWS that

*No discontinuity was detected at the year 2002 indicating good adjustment procedures from Human Observer to instrument at the transition time.*

Lines 568-570: How do you explain this result?

Author's response: The increased in cloud cover together with a decrease in cloud optical thickness is an interesting result. The cause is unclear and contrary to [our] intuition. We could speculate on this issue but that would detract from the main results. However it is a valid point of attention so we included:

The implication is that clouds have become (optically thinner) but at the same time more frequent, the cause of which is unclear.

Technical corrections:

Check the reference at line 77;

Author's response: Yes, corrected in the references

Line 105: It is the first time that the abbreviation ACI is used in the text so it is necessary to define it (even if it is already defined in the abstract);

Author's response: yes, was done

Line 226: Define all the parameters in equation 16;

Author's response: yes, was done, but will be part of the newly formed appendix.

Some additional references are necessary, for example at lines:

298,

Author's response: this is standard definition of optical thickness, we believe that this does not need a reference.

303,

Author's response: this is the Mean Value Theorem, which we included in the text.

315,

Author's response: this is indeed an imprecise statement. We used the average value over the Netherlands based on the ERA data. So we changed in the text:

*...and a value of 1000 m was used to reflect conditions over the Netherlands.*

323,

Author's response: yes, this is Boers (1994), and included in the reference list

350;

Author's response: yes, this is Twomey (1977) and a huge number of others!

Check the reference at lines 584 and 676.

Author's response: yes it is Sanchez-Lorenzo, not the other way around. We corrected.

---

## Author Comment (AC2) · 8 May 2017

**Authors response to Anonymous Referee #2**

Second review of "Impact of aerosols and clouds on decadal trends in all-sky solar radiation over the Netherlands (1966-2015)," by R. Boers, T, Brandsma, and A. P. Siebesma

General comments:

This paper shows an innovative approach to isolate the sources of dimming and brightening for the Netherlands over a 50-year period. It is of high scientific significance because the trends of dimming then brightening over the period of study are well known global phenomena, but their causes are not universally consonant. Problems in documenting dimming, especially, in the early part of the period of study are hampered by a lack of appropriate data that have been generally available during the recent brightening period. In my opinion, the authors have ably used available sources of data to create credible proxies and correct data appropriately (e.g., cloud fraction over the transition from human observers to ceilometers) to study dimming and brightening over the Netherlands and impressively determined the relative contributions of aerosols and clouds to those phenomena.

We thank the Referee for the comments. Below follow the comments and our answers to them:

The mathematics used is impressive but cumbersome and could be simplified for the reader by including only the final equations in the main text of the paper, with complete descriptions of their components and the details of the derivations placed in an appendix. That is only a suggestion. In addition, more frequent reminders to the reader of the time periods that terms discussed represent would be useful.

Authors response: This point was also brought up by Referee #1. The authors have decided to put the Method sections 2.1, 2.2, 2.3, and 2.4 in an appendix and only write down the end result (including a description of it) of the final equations, namely Eq. (17) and Eq. (21). This is a substantial reduction in the main text which improves the flow of the manuscript.

One major concern is that the primary results presented in Table 2, and summarized in the abstract, are not intuitive and require more explanation. That is, the 50-year trends in clear-sky and cloud-base radiation are both greater than the 50-year trend in all-sky radiation. Intuitively, the former two trends should add to the all-sky trend. However, when examining sub-trends during the periods of dimming and brightening separately, which can be done from the results presented in Table 2, the component trends (i.e, clear-sky and cloud-base) do sum better to the reported all-sky trends for those sub-periods, within the margins of error presented.

Authors response: For the better part of a year the first author shared with Referee #2 the notion that the clear-sky and cloud-base radiation trends should add up to the all-sky trend. This is also what is implicitly stated in many studies on this subject. So we can appreciate Referee #2 's difficulty in understanding the issue at hand. It was only after the third author was able to write down the equations on the separation of clear and cloudy signals that we realized how wrong we were. Intuition is not a good guide in this matter. Perhaps the simplest way to see it is to picture the 'all-sky' as partly 'clear-sky' and partly 'cloudy-sky' . This then must mean that both should be weighted by their sky fraction, and that changing the sky fraction itself should also be of importance in determining the trend. Furthermore, included in the complexity associated with understanding this analysis are the concepts associated with the separation between 'real' data and its 'proxy' counterpart. These points of Referee #2 are quite valid in our opinion and additional statements explained the results should be of assistance in understanding all of these issues, so we included at the end of the discussion of Table 2 on the proxy analysis:

I

*Note that both the trends in all-sky radiation and the trend in all-sky proxy radiation are given in the table. The trend in all-sky radiation is simply inferred from the data whereas the trend in all-sky proxy radiation is computed from Eq. (3). Thus, contrary to common notion the trend in measured all-sky radiation cannot be recovered from the trends in proxy data. It is only the all-sky proxy trend that can be recovered from the clear-sky proxy term and the cloud-base proxy term of Eq. (3) and in addition from the fractional cloudiness term of Eq. (3). Note furthermore that the fractional cloudiness term in Eq. (3) is a scaled version of the trend in fractional cloudiness, whereas the other two are scaled versions of the trend in clear-sky proxy radiation and cloud-base proxy radiation.*

Also at the end of the appendix when the trend analysis is explained we included some statements on the counterintuitive aspects of understanding the trends:

*The implications of this expression are quite important. Eq. (A22) demonstrates that the trend in all-sky radiation is not a simple summation of trends in clear-sky and cloudy-sky trends, which would perhaps be an intuitive notion when seeking to explain the observed trend in all sky radiation. Eq. (A22) demonstrates that a) the trends in clear-sky and cloud-base radiation need to be weighted by their fractional occurrence in the atmosphere, and that b) there is a third term constituting the trend in fractional cloudiness scaled by the difference in average cloud-base and clear-sky radiation. Furthermore, the additional fourth term, which is shown to be negligible in the current analysis, may not always be small when there are significant cross correlations between the perturbations.*

One potential problem I see in your analysis is in your interpretation of eq. 21. As stated on lines 256-257, the over-bars in the equation represent 50-year means and the primed quantities represent yearly deviations from decadal averages. This inconsistency may lead to problems. Another potential problem I see is that the weights applied to the clear-sky term (.32) and the cloud fraction term (.68) represent the fractional periods of clear and cloudy conditions over the entire 50-year period. Since those fractions likely change through the 50-year period, I believe it would be beneficial to analyze eq. 21 over decadal periods, using decadal means, yearly deviations from decadal means, and decadal weights of fractional clear-sky and cloudy periods to compute $S'(y_k)$. Then, the means and deviations used would be internally consistent and the fractional mean periods of clear and cloudy skies would be appropriate to the decade being analyzed.

Authors response: This was an error in the text: it should read: 'yearly deviations from an average over 5 decades of the yearly averages'. We regret the confusion caused by this error. Given Referee #2 's interpretation of the erroneous text in the original manuscript the suggestions given are quite logical. However, the two time periods given in the text (1966 – 1984) and (1984 – 2015) are not multiples of a decade which impedes the analysis suggested here (the break point of 1984 would have to be changed). We prefer to leave as is (but of course corrected the error in the text which now appears in the appendix).

Specific comments:

Abstract:        As detailed in the general comments, the differences among the three trends listed needs more explanation.

Authors response: Yes, the problem with the abstract is that the aspect of real data versus proxy data were not even addressed, so this point certainly needs to be amplified: We inserted a set of statements in the abstract more fully explaining what we did, which now reads as follows:

*A 50-year hourly dataset of global shortwave radiation, cloudiness and visibility over the Netherlands was used to quantify the contribution of aerosols and clouds to the trend in yearly-averaged all-sky radiation ($1.81\pm1.07\ Wm^{-2}$/decade). Yearly averaged clear-sky and cloud-base radiation data show large year-to-year fluctuations caused by yearly changes in the occurrence of clear and cloudy periods and cannot be used for trend analysis. Therefore, proxy clear-sky and cloud-base radiations were computed. In a proxy analysis hourly radiation data falling within a fractional cloudiness value are fitted by monotonic increasing functions of solar zenith angle and summed over all zenith angles occurring in a single year to produce an average. Stable trends can then be computed from the proxy radiation data A functional expression is derived whereby the trend in (proxy) all-sky radiation is a linear combination of trends in fractional cloudiness, (proxy clear-sky radiation and proxy cloud-base radiation. Trends (per decade) in fractional cloudiness, proxy) clear-sky and (proxy) cloud-base radiation were respectively $0.0097\pm0.0062$, $2.78\pm0.50\ Wm^{-2}$, and $3.43\pm1.17\ Wm^{-2}$. To add up to the (proxy) all-sky radiation the three trends have weight factors, namely the difference between the mean cloud-base and clear-sky radiation, the clear-sky factor (1-fractional cloudiness) and the fractional cloudiness, respectively. Our analysis clearly demonstrates that all three components contribute significantly to the observed trend in all-sky radiation. Radiative transfer calculations using the aerosol optical thickness derived from visibility observations indicate that Aerosol Radiation Interaction (ARI) is a strong candidate to explain the upward trend in the clear-sky radiation. Aerosol Cloud Interaction (ACI) may have some impact on cloud-base radiation, but it is suggested that decadal changes in cloud thickness and synoptic scale changes in cloud amount also play an important role.*

l. 28-30  Brightening in the U.S. from the mid 1990s to ~2011 is attributed primarily to a reduction in cloud cover in Long et al. (2009), Augustine and Dutton (2013), and their results are based on data alone.

Authors response: Long et al is equivocal about it, certainly in comparison to Augustine and Dutton. At this point in the text it is inappropriate to include these references because this paragraph deals with European data. We inserted the references at the end of the next paragraph though where more general comment are made.

Why mention that climate models are not capable of reproducing these trends. Are climate models even capable of resolving the cloud physics necessary to resolve the various cloud types and cloud cover responsible for dimming and brightening? I doubt it.

Authors response: On balance a fair point: The paper hardly discusses models so we remove the reference to Allen.

l. 36-42  Wang et al. (2013) has clearly shown that errors associated with single black detector pyranometer measurements adversely affect trends in solar radiation. In this respect, GEBA data are not of unmistakable quality.

Authors response: we adjusted the sentence to:

*GEBA data can be used to good effect because of the fact that many stations have submitted data, but the peculiarities of the radiative signals typical to individual localities are invariably lost in the abundance of data.*

l. 73-73  Does your statement that cloud cover data are collected simultaneously with radiation data apply to the Netherlands? That may not be true for most of the radiation stations over the globe.

Authors response: True enough, in that sense the Netherlands may indeed be the

exception. The statement was adjusted to:

*Also, data on fractional cloudiness needs to be collected simultaneously with radiation data.*

l. 105          Define acronyms

Authors response: Done

l. 132-133       How is a representative cosine of the solar zenith angle (SZA) determined
                for a particular hour? Do you use the cos(SZA) at the midpoint of the hour or
                do you average the SZAs within the hour period? Averaging SZAs at low sun
                does not provide a good "representative" SZA for the hour.

Authors response: We use mid-point of the hour which is now stated in the text. [in the
                appendix]

l. 199          Change "to suggesting that" to "that suggests"

Authors response: Done

l. 220-221       To get proxy data for $S_{cj}(y_k)$ I assume that you plot the data corresponding
                to the variables in eq. 15 in a way analogous to Langley plots, and then use
                the resulting relationship to generate the proxy data per okta. Correct? If so, I
                assume that the scatter, and thus uncertainty, of those plots would be small
                for clear-sky data and larger for oktas 1 through 8. Those uncertainties would
                define the error in the various terms of eq. 21. Were those uncertainties
                incorporated into your analysis? They should at least be presented in some
                form—if possible. If my interpretation of your proxy data generation is wrong,
                please better explain your method in the paper.

Authors response:

Yes, thank you for bringing this point to the fore. Scattering and thus uncertainty is larger for
                higher fractional coverage. We include that in the text (see below). The rest
                of Ref #2's statement is a rather complicated point though, as for the trend
                analysis we mix uncertainties in the data points with natural atmospheric
                variability on a multi-year time scale. The M-K test does not separate these;
                it simply uses the given data to derive trends. So it will be hard to satisfy Ref
                #2 's comments completely. We experimented with other types of trend
                analysis than the M-K including one whereby the trend was calculated by
                randomly imposing errors of a specified value on the data points and
                recalculating the trends and thus their uncertainties. We got very similar
                results as our M-K analysis. As the M-K analysis is a standard test of
                significance used in this type of study we prefer to keep it in (see for example
                Long et al, 2009). Nevertheless it may be of use to clarify certain parts of the
                text. Therefore:

 We included in the appendix several extra statement explaining the function B and the
                errors involved:

*The parameters α and β are constants determined by fitting the data. The method expressed in
                Eq (A18) is equivalent to the Langley method of obtained optical thickness with
                the only difference the weak dependence of B on sun angle. Such dependence
                is necessary to include because the diffuse radiation arriving at the surface is
                weakly dependent upon $\mu_0$,*

And a little lower in the appendix:

*Eq (A19) expresses the dependence of atmospheric optical thickness on $\mu_0$. Regression fits using Eq. (A19) carries uncertainties into the parameter B and through Eq. (A18) into parameter G and into Eq. (A20). For clear-sky the scatter is small but for skies under (partly) cloudy skies the scatter is larger. The standard 1-sigma uncertainty associated with the clear sky proxy computed in Eq. (A20) is 2-3%, increasing to 8-9 % for high okta values.*

And in the main text

*Tests of trends will be performed using the standard Mann-Kendall (M-K) (Kendall, 1975) non-parametric test often used in this type of analysis (see f.e. Long et al., 2009). after the time series was first decorrelated. The uncertainty value attached to the trend is a test of significance indicating the 95% confidence interval of the calculated slope line. The uncertainties in trend are due to two factors, namely those in yearly-averaged values of $S_p$ as a result of uncertainties in fitting constants in Eq. (A19) (see Appendix for details) and due to natural variability of a multi-year or even decadal origin. Thus the stated uncertainty in output trend is a mix of both factors.*

l. 249          This sentence needs to be reworded.

Authors response: 'Is' was changed into 'it' which clarifies it!

l. 270          The 4[th] term is shown.

Authors response: Point was also noted by Referee #1. It was remnant of a previous version of this manuscript. It is now corrected.

l. 321          "is" should be changed to "in"

Authors response: Yes, done

l. 355          Since you are discussing aerosol optical depth and cloud optical depth in the same sentence, references to each should be specific. In this case, insert "cloud" in front of "optical depth" on this line.

Authors response: Yes, done

l. 366-368    By "cloud fractional coverage at specific cloud cover," do you mean the transformation of okta observations to fractional cloud cover? If so, please state this more clearly.

Authors response: An imprecise statement on our part. We corrected:

*In our analysis, cloud fraction is obtained in a straightforward manner by counting the hourly cloud data so that the hypothesis that changes in aerosol results in changes in cloud cover can be tested.*

l. 414    The sentence beginning with "The SNHT was applied …" is difficult to understand. What do you mean by "reduced with?" and how does that apply to the a) and b) permutations?

Authors response: Yes, we agree, it was unclear and we changed it:

*In the first test each station series was subtracted by the mean of the four other station time series. In the second test each station series was subtracted by the other four station time series separately.*

l. 516    What is ERA?

Authors response: An acronym in an acronym, not common but unfortunately not unheard of either in the atmospheric sciences! We expanded it:

*the European Centre for Medium Range Weather Forecast's Re-Analysis project, ERA*

l. 537    "b)" should be changed to "2)"

Authors response: Done

l. 555    The sentence beginning with "Cloud amount is increasing…" is counterintuitive. It would benefit by inserting "in solar radiation at the surface" after "overall trend."

Authors response: Done

l. 560    The large tick marks represent 10 Wm$^{-2}$.

Authors response: Done

---

## Author Response (AR2)

**Authors response to Co-Editor's notes**

**Comments to the Author:**

The paper presents a very interesting analysis on solar radiation measurements at the Netherlands. The use of these long term time series together with the mathematical approach makes the paper very interesting and suitable for publication in ACP.

However, the "innovative" point which is the proxy approach on presenting and interpreting the data could be also a disadvantage for the paper quality if some more analysis is not presented. Mainly comparing the actual measurement results with the proxy results.

For example right now it is mentioned that the all sky measurements and proxy all sky computations agree within 5% but also all the results shown (e.g. in figure 4) point out to (50 year) changes of that ( $\sim$ 5%) order.

**Authors response:** Thank you for your review, it was quite helpful in clarifying some additional issues with respect to proxy versus real data. We included an extra figure showing the proxy versus real data and thus had to renumb er all figures. We had in fact placed this figure in one of the earliest versions of the manuscript, but the Figure is rather difficult to interpret because it opens a discussion to more subtle aspects of seasonal trends. Such a discussion we had wished to avoid as the concept of a proxy is already difficult enough to understand anyway. Also a full discussion on seasonal trends is beyond the scope of the paper (but certainly interesting to study at some time).

Nevertheless on rereading the manuscript there is a logical place to put it, namely between Figure 3 and Figure 4 (see earlier revised version) as a natural transition between 'real' and 'proxy' data. However, it requires substantial discussion the content of which you find below. On balance it probably assists the reader in appreciating the differences between proxy and real data.

More specific comments.

**abstract and table 2:**

The Trends (per decade) in fractional cloudiness (abstract and table) and Fractional cloudiness term of Equation 3 (table) have to be clarified. This is because in the description of data used is not clear if that is octas (it is not but it is not clear), or a number from 0 to 1 (thus this 0.0097 can be translated to a 0.97% per decade change at least in the abstract). In another section this is expressed as cloud cover percentages as a function of cloudiness based on the 2010 Boers paper and then grouping the 0-1 and 7-8 octas conditions. So in the metadata sections the way that this fractional cloudiness is calculated could be more clear.

**Authors response**: The mixed use of fraction and percentage is indeed confusing. We amended the text in section 3.4 to include an explicit mentioning of our separate usages of okta and fractional and changed the reference to cloud fractions in percentage to cloud fractions:

We adhere to the results of their study (their section 2.3, table 1) where for okta 0-8 the following cloud fractions are given: 0.00, 0.0615, 0.2494, 0.3751, 0.5003, 0.6256, 0.7518, 0.9507, 1.00.

**And later on:**

In this paper we use both the terms cloud fraction and okta. When selecting radiation values for a particular okta value (index j in section 2), the cloud fraction attributable to that particular okta value (i.e.  $c_j$  in Eq. (2)) is used to compute the (proxy) radiation. The computation of the yearly mean fractional cloudiness (with index k),  $c_k$  as per Eq. (A4) simply takes the average over all  $c_j$  values occurring over the entire range of okta values.

Proxy functions and data.

The fit of hourly values of a certain octa value with solar zenith angles.

So for each of these points you are using a cloud observation (or measurement after 2002) and a solar zenith angle. Since solar zenith angles are not linearly changing with time, you have to clarify which one is it used. In addition, one hour solar radiation averages attributed in one cloud observation could end up in quite complex "shapes" of these fits. Especially for 3-6 octas where more or less the data points within a certain octa and a certain solar zenith angle could be easily a bimodal distribution based on the sun obscureness by clouds or not. Could you add some comments on this issue and on the proxy functions retrieval ?

**Authors response**: This conception is probably due to lack of clarification on our part of the use of the function G in the appendix. We amended the text there (section (A1.2) by elaborating a bit more on what it means to use a smooth function with the complete marginal distribution of sun angles. This hopefully assists in clarifying this point.

The year-to-year determination of proxies in Eq. (A20) is used in this paper as it will yield more stable results than the determination of 'true' averages. The fitted functions G in (A18) are smooth, monotonic increasing functions for all okta values. Their use together with the marginal distribution (i.e. all 8760/8784) of zenith angles to compute the proxy will avoid all seasonal, yearly or multi-year variations that are inherent in the application of the distribution  $f_k(\mu_{0i}|c_i)$  for which the yearly variable numbers of  $\mu_{0i}$  values necessary to compute the conditionally sampled data are used. Therefore, the computed trends of proxies will reflect the yearly changing transmission through the atmosphere rather than the more spurious effects of random or multi-year seasonal variability in radiation at various cloud fractions.

I think it would be necessary to show a figure of actual (all sky) measurements and proxy (all sky data) as a function of time. This in order to provide a link on your study and other (most) studies that interpret their results using actual measurements. Also in order to quantify the results based on the fact that percentage changes of the 50 year dataset are in the same magnitude with proxy and actual measurement yearly averages.

**Authors response**: As mentioned a new figure was added, but the figure requires substantial additional discussion as it is full of subtle points that demand explanation. In fact it opens a box (neither a can of worms nor Pandor a's box!) to a much deeper interpretation of the results because there is a wealth of additional information on seasonal changes buried in the data. During analysis we went quite some way to uncover seasonality issues but treatment of this issue is far beyond the scope of this work which only deals with yearly changes. Therefore we only hint on seasonal issues:

Figure 4 shows the all-sky radiation and its proxy. Difference between the two averaged over the 50 years is 4.34%. However there is some year-to-year variability. For example in the years just prior to 2000 the differences are less than 1%, while in the period 2012 – 2015 it is about 8%. Such variations are the result of a) natural year-to-year variations in the distributions of zenith angles attributable to the individual okta values for the all-sky radiation, b) possible more systematic changes to the distribution of sun angles per okta value (i.e. seasonal changes on a multi-year time scale) and c) the uncertainties in the line fits necessary to compute the proxy radiation. At any rate there appears to be a systematic bias between the two time series of 4 Wm-2. This is primarily caused by the fact that in the Netherlands mostly cloudless skies occur in summer months when the sun is high in the sky. This means that when the proxy radiation is computed using the marginal distribution of sun angles (see Appendix) there will be an inevitable shift towards lower sun angles (i.e. smaller radiation values) in comparison to the real flux for which the conditional distribution of sun angles is used in its computation. This situation is peculiar to the Netherlands and is unlikely to be a universally observable feature. Because of these differences there will also be some differences between the trend values of the real (observable) and proxy (calculated) fluxes to be calculated later on (see later in Table 2).

And later under Table 2 another set of sentences is inserted:

In comparing the trends between all-sky and all-sky proxy flux we find that for the period 1966 – 2015 both trends are almost the same. For the period 1966-1984 they are different in sign (but neither trend is deemed significant). For the period 1984 – 2015 both are significantly positive but the trends differ by 30% from each other. As explained above (see under Figure 4) such differences are to be expected due to the fact that the method to calculate the proxy all-sky flux uses the marginal distribution while the calculation of the all-sky radiation uses inherently the conditional distributions of sun angles, which can exhibit year-to-year variations or multi-year seasonal changes.

I miss something about the instrument calibration. Since we are talking about instruments starting operating in the 60's a comment on the instrument uncertainty should be included especially in order to be compared with the actual solar radiation changes that are presented for the 50 year period.

**Authors response**: we included some more details on the calibration and the impact of uncertainty of hourly radiation data on the yearly averages (see section 3.2):

When proper calibration procedures were eventually in place instruments were rotated from the instrument pool on a 12 – 15 month cycle out of KNMI, where calibration was done according to fixed procedures. Based on these procedures individual hourly observations were estimated to have a random (i.e. unbiased) uncertainty of 8%. However, per year 8760 / 8784 are used to produce the average and with no bias in individual observations the uncertainty in the average is negligible in comparison to the trends in radiation to be shown below (see Section 4).

Figure 4 is very interesting and the results coming from it are also unique. Especially the fact that even if fractural cloudiness/cloud fraction term, increases/decreases the all sky proxy term is increasing due to the cloud base term or thinner clouds.

My question for this figure is:

Since all sky proxies are calculated (and more or less represent the actual measurements), plus the cloud fraction term is calculated based on (independent than the pyranometers) cloud observation, plus the clear sky term is also based on the G (umbrella function); could it be that the cloud base term is simply the term (all sky proxy) – (clear sky proxy) – (cloud fraction term)? To try to be more clear: if someone would have to re-evaluate the cloud observation data in the future, would that affect also the cloud base term ? (since all sky term is radiation and in principle it will not change and the clear sky term will also not change much).

**Authors response**: The co-authors remarks are factually correct that the sum of the terms should add up to the proxy all-sky radiation within the margins of error stated. This can be easily verified from the tables: They all fall within the margin of error indicated by the M-K fitting procedure (as they should). But this was already noted by one of the referees. The fact that they are not exactly the same is inevitable because the fitting procedure always introduces some small changes.

However, it should again be stresses that it is the sum of the components that make up the all-sky proxy, and not the other way around as the co-editor is suggesting. The all-sky proxy cannot be obtained directly from an analysis of the real data. It can only be inferred from the analysis of proxy radiations under clear, partly cloudy and cloudy skies. In the event that the proxy and real all-sky radiation turn out to be exactly the same, then this must mean that the conditional distributions are exactly the same as the marginal distribution of sun angles. This is highly unlikely to ever occur in the real world.

The proximity of the real and proxy all-sky radiation could serve as a test of whether the proxy radiation analysis is in fact done correctly. Consequently, if another fitting procedure to get the clear / cloudy sky proxy is used (when an investigator decides on another method of calculation), the sum of the terms in Eq. 3 should always end up close to the all-sky proxy, and any differences between them should have a reasonable explanation. As mentioned, for the Netherlands the observed bias is caused by the bias towards high sun angles for [almost] clear skies that heavily weight the real all-sky data towards high values in comparison to the proxy. This is explained in the text. But this may not always be the case for other locations.

This point does not, in our opinion, warrant another change in the text.

[revised manuscript text omitted]